# The impact of mesh size and microphysics scheme on the representation of mid-level clouds in the ICON model in hilly and complex terrain

Nadja Omanovic[1,*], Brigitta Goger[2,*], and Ulrike Lohmann[1]

[1]Institute for Atmospheric and Climate Science, ETH Zurich, Zurich, Switzerland
[2]Center for Climate Systems Modeling (C2SM), ETH Zurich, Zurich, Switzerland
[*]These authors contributed equally to this work.

**Correspondence:** Nadja Omanovic (nadja.omanovic@env.ethz.ch) and Brigitta Goger (brigitta.goger@c2sm.ethz.ch)

**Abstract.** The rise in computational power in recent years enables researches and national weather services to conduct high-resolution simulations down to the kilometric ($\Delta x = \mathcal{O}(1\,\mathrm{km})$) and even to hectometric ($\Delta x = \mathcal{O}(100\,\mathrm{m})$) scale for both weather and climate applications. We investigate with the state-of-the-art numerical weather prediction model ICON how mid-level clouds are represented on a mesh size of $1\,\mathrm{km}$ and $65\,\mathrm{m}$, respectively, and for two bulk microphysics schemes, one-moment and two-moment cloud microphysics. For this analysis, we leverage the abundant observational data from two independent field campaigns in Switzerland (CLOUDLAB, hilly terrain) and Austria (CROSSINN, complex terrain). With four case studies, we show that while the temperature fields around the campaign sites are well represented in both mesh sizes, the $65\,\mathrm{m}$ resolution simulates a more realistic vertical velocity structure beneficial for cloud formation. Therefore, the largest differences for the representation of clouds lies in the two mesh sizes: The $1\,\mathrm{km}$ simulation in hilly terrain does not capture the observed clouds in both cloud microphysics schemes. Here, the higher resolution of the vertical velocities in the $65\,\mathrm{m}$ proves to be crucial for representing the investigated cloud types, and the two-moment microphysics scheme in general performs better with respect to the cloud characteristics because it considers variations in cloud droplet and ice crystal number concentrations. In complex terrain, the differences between the mesh sizes and the cloud microphysics schemes are surprisingly less, but the $65\,\mathrm{m}$ simulations with two-moment cloud microphysics shows the most realistic cloud representation.

## 1 Introduction

Numerical weather prediction (NWP) models have undergone immense improvements in the last decades due to the rise of computational power (Bauer et al., 2015; Palmer, 2017): Operational NWP forecasts nowadays run at the kilometric range at various European weather services (e.g., MeteoSwiss with ICON at $\Delta x = 1\,\mathrm{km}$[1], UK Met Office with UM at $\Delta x = 1.5\,\mathrm{km}$[2], Météo France with AROME at $\Delta x = 1.25\,\mathrm{km}$[3]), and kilometer-scale climate models with their high-resolution output fields (e.

---

[1]https://www.meteoswiss.admin.ch/about-us/research-and-cooperation/projects/2023/icon-22.html, last access: October 31, 2024

[2]https://www.metoffice.gov.uk/research/approach/modelling-systems/unified-model/weather-forecasting, last access: October 31, 2024

[3]https://donneespubliques.meteofrance.fr/?fond=produit&id_produit=131&id_rubrique=51, last access: October 31, 2024

g., precipitation) become more and more common for regional (e.g., Ban et al., 2014; Leutwyler et al., 2016) as well as global simulations (Schär et al., 2020; Hohenegger et al., 2023).

One of the major advantages of kilometric simulations ($\Delta x = \mathcal{O}(1\,\mathrm{km})$) is the more realistic representation of model topography in the domains, allowing for more detailed terrain-induced circulations in models, such as the thermally-induced valley wind system (Schmidli et al., 2018; Goger et al., 2018, 2019; Heim et al., 2020; Mikkola et al., 2023). Another advantage of kilometric simulations compared to coarser mesh sizes ($\Delta x = \mathcal{O}(10\,\mathrm{km})$) is that the mass flux parameterization can be switched off, because deep convection is already resolved on the grid (Chow et al., 2019). Hentgen et al. (2019) point out the improved representation of clouds in kilometric simulations over Europe, and more recent studies even suggest to go towards the hecto-metric range ($\Delta x = \mathcal{O}(100\,\mathrm{m})$), further improving cloud representation in numerical models (Stevens et al., 2020; Miyamoto et al., 2013). Heinze et al. (2017a) noted in their simulations at $\Delta x = 100\,\mathrm{m}$ over Germany a more detailed representation of cloud patterns over Germany, and Schemann et al. (2020) compared large-eddy simulations (LES) to local observations to find the best representation of clouds in the model in small domains with realistic mesoscale forcing. Therefore, LES are favorable for process studies, also because local circulations and the boundary layer structure affecting cloud formation are represented well, if realistic atmospheric forcing (e.g., from kilometric NWP model runs) and high-quality surface parameter datasets are used (e.g., Heinze et al., 2017b; Gerber et al., 2018; Umek et al., 2021; Goger et al., 2022; Rohanizadegan et al., 2023; Goger and Dipankar, 2024; Voordendag et al., 2024).

Still, there are limitations to the ability to interpret LES results given that many cloud processes act on a sub-micron scale, and thus still need to be parameterized. Nevertheless, LES proved to be a useful tool for investigating, e.g., marine boundary layer clouds given their inadequate representation in global models (e.g., Stevens and Bretherton, 1999; Nam et al., 2012). This includes studies ranging from investigating the effect of dynamics, such as entrainment (e.g., Siebesma et al., 2003; Duynkerke et al., 2004; Bretherton et al., 2007; Sandu and Stevens, 2011; Bretherton and Blossey, 2017; Jeong et al., 2023) to aerosol-cloud interactions (e.g., Jiang et al., 2002; Xue et al., 2008; Sandu et al., 2008; Andrejczuk et al., 2010; Twohy et al., 2013; Tonttila et al., 2017; Atlas et al., 2020; Diamond et al., 2022; Delbeke et al., 2023; Li et al., 2024; Perez et al., 2024). Often these studies included comparisons to observational data gathered during campaigns targeting marine boundary layer clouds (e.g., Roberts et al., 2010; Allen et al., 2011; Schulze et al., 2020; Wang et al., 2022; Howes et al., 2023), which also lead to model improvements with respect to the formulation of parameterizations and numerics (e.g., Stevens et al., 1996; Stevens and Bretherton, 1999; Yamaguchi and Feingold, 2012; Pressel et al., 2017; Mellado et al., 2018). LES studies on clouds over land include the already mentioned model evaluation of ICON in large-eddy mode over Germany (Heinze et al., 2017a; Schemann et al., 2020), the evaluation of shallow cumulus clouds over the Great Plains (Zhang et al., 2017), the more idealized approach of evaluating either the impact of turbulence on moist convection (Strauss et al., 2019), the dependence of convection and precipitation on grid spacing (Moseley et al., 2020; Singh et al., 2021), or the formation of clouds over mountainous terrain (Panosetti et al., 2016). All these studies highlight the advantages of LES for studying cloud processes.

When focusing on clouds in models, the question arises which level of complexity is needed to "properly" represent them in terms of parameterizations of clouds. The answer to this question is certainly constrained by the available cloud microphysics schemes in the model but also by the available computing resources. The more complex schemes, which are supposed to be

more accurate, also require more computing resources for resolving more processes for cloud formation and evolution. The simplest cloud microphysics schemes are so-called bulk cloud microphysics schemes, which specify a selected number of hydrometeor classes (e.g., cloud droplets, ice crystals, snow, rain, graupel, and hail) and directly predict the mass mixing ratios (one-moment cloud microphysics, 1M) or in addition the number mixing ratios (two-moment cloud microphysics, 2M) of these hydrometeors. This, however, requires the parameterization of shapes and size distributions of the prognostic particles (Doms et al., 2021). The prediction of both mass and number mixing ratios already results in a large increase in required computing resources such that national weather services, often perform their operational forecasts with an one-moment cloud microphysics scheme (Buzzi, 2008; Doms et al., 2021). More advanced cloud microphysics schemes include spectral bin microphysics (e.g., Simmel et al., 2002; Khain et al., 2011) or so-called Lagrangian superparticles (e.g., Andrejczuk et al., 2008, 2010; Shima et al., 2009; Hoffmann, 2016). The wide range of possible cloud microphysics schemes also inspired several studies comparing bulk and bin cloud microphysics schemes (e.g., Endo et al., 2015; Sato et al., 2015; Zhang et al., 2017; Witte et al., 2022), Eulerian and Lagrangian frameworks (e.g., Grabowski, 2020), or 1M versus 2M. For the latter, studies pointed out the better performance of 2M highlighting the potential for better representing clouds in model simulations (e.g. Baldauf et al., 2011; Bryan and Morrison, 2012; Van Weverberg et al., 2014; Kovačević and Ćurić, 2015; Kondo et al., 2021).

In this study, we want to answer the question if the two-moment microphysics scheme (2M) is better suited for representing and also studying clouds compared to the one-moment microphysics scheme (1M). We further expand this question by also looking at the dependence of horizontal resolution ($\Delta x = 1\,\mathrm{km}$ and $\Delta x = 65\,\mathrm{m}$) and the underlying topography (hilly and complex terrain). The evaluation is based on case studies and model-observation comparisons utilizing high-resolution remote sensing observational data from two field campaigns in Switzerland and Austria, respectively. We focus on mid-level clouds with either a stratiform or a more convective character. This work should help to understand the differences between two bulk microphysics schemes, their benefits and disadvantages and to highlight limitations in process representation on various grid scales and above different terrains. We first provide an overview of the field sites, campaigns, and the model setup (Sect. 2) before discussing the model performance for hilly terrain (Sect. 3.1) and complex terrain (Sect. 3.2). After a comparison of the case studies and the discussion (Sect. 3.3), we highlight our main findings in the conclusions (Sect. 4).

## 2 Field campaigns and model setup

In the following, we describe the field campaigns and model setup used to conduct our study. The two field campaigns were conducted independently of each other and with different research foci (cloud and boundary layer research, respectively) at two different locations (hilly terrain in the Swiss Alpine foreland and highly complex terrain in the Austrian Alps, respectively). We took advantage of the availability of observational data obtained from a broad range of instruments employed in both campaigns to validate our model. Table 1 gives an overview of the field campaigns, the instruments used in this study, and the cases simulated here.

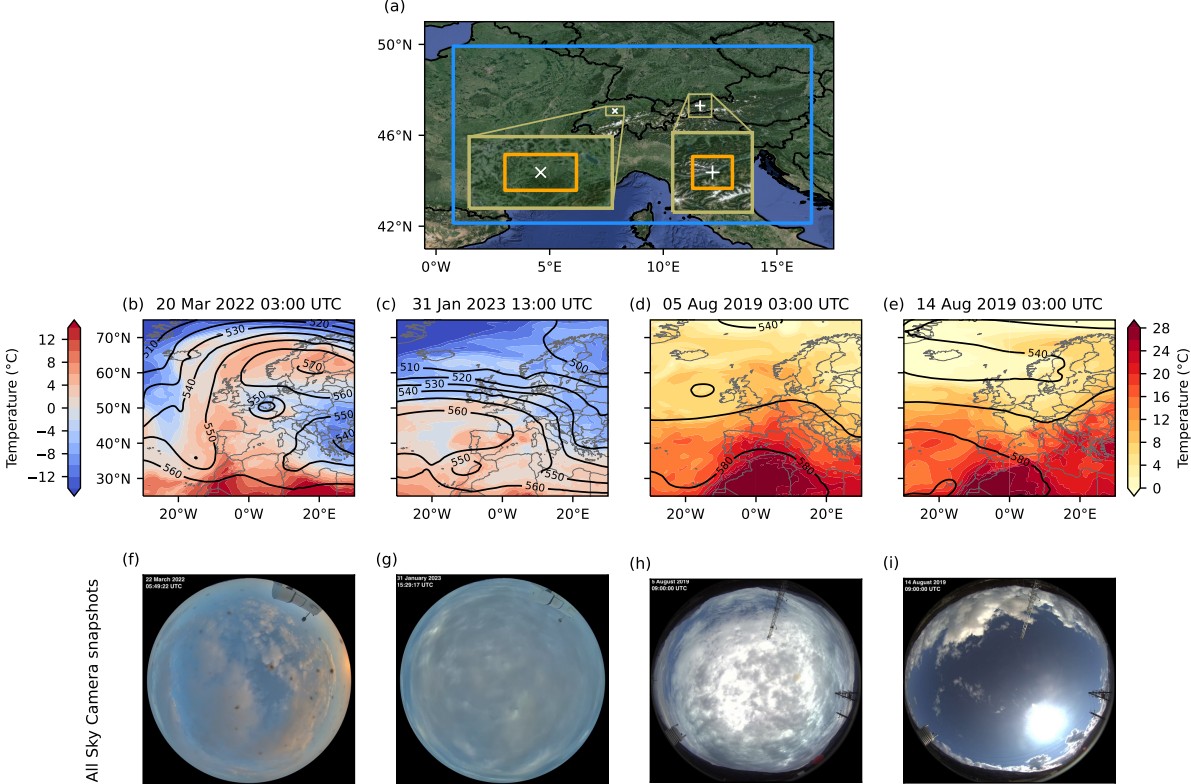

**Figure 1.** First row **(a)** shows the model setup with "x" and "+" denoting the field sites of CLOUDLAB and CROSSINN, respectively. The two model domains of interest in this study are the blue ($\Delta x = 1\,\text{km}$) and the orange boxes ($\Delta x = 65\,\text{m}$), with the olive-green box ($\Delta x = 125\,\text{m}$) denoting the domain in which the $65\,\text{m}$ simulation was nested in. Map taken from Google satellite images (©Google Maps). Second row **(b-e)** shows the large-scale weather situation for our four case studies based on ERA5 data (Hersbach et al., 2020) with the coloring representing the temperature (°C) at $850\,\text{hPa}$ and the black contours showing the geopotential height (decameter) at $500\,\text{hPa}$. Last row (**(f)-(i)**) is a collection of snapshots from the Allsky cameras located at the two field sites, where each snapshot corresponds to one case study and shows a typical cloud situation of that day (time stamp in upper left corner). Panels **(b)**, **(c)**, **(f)**, and **(g)** (left side) are the case studies for the CLOUDLAB field site, while the others (right side) are for the CROSSINN field site. Panel **(f)**: dirt fragments (dark dots) were present on the Allsky camera lens.

## 2.1 Field campaigns

### 2.1.1 CLOUDLAB

The CLOUDLAB project used glaciogenic cloud seeding to study ice processes in supercooled low stratus clouds (Henneberger et al., 2023; Miller et al., 2024; Omanovic et al., 2024a). The field site is located at the edge of the Swiss Plateau on a prominent hill at $920\,\text{m}$ a.m.s.l. surrounded by hilly terrain. It entailed three field campaigns (January 2022 - March 2022, December 2022

**Table 1.** Overview of field campaigns including their respective coordinates, altitude, duration, instruments used in this study, and the selected case studies.

| Field campaigns | CLOUDLAB | CROSSINN |
|---|---|---|
| Coordinates | 47°04'14"N, 7°52'22"E | 47°18'19"N, 11°37'19"E |
| Altitude | 920 m a.m.s.l. | 545 m a.m.s.l. |
| Duration | Jan 22 - Mar 22<br>Dec 22 - Feb 23<br>Dec 23 - Feb 24 | Jul 19 - Oct 19 |
| Instruments | Allsky camera<br>Ceilometer<br>Cloudnet<br>(-)<br>Microwave radiometer<br>(-) | Allsky camera<br>Ceilometer<br>(-)<br>LIDAR<br>Microwave radiometer<br>Radiosondes |
| Case dates | 20 March 2022<br>31 January 2023 | 5 August 2019<br>14 August 2019 |

- February 2023, and December 2023 - February 2024), during which an extensive setup of remote sensing and meteorological instruments were present. A full list of the instruments can be found in Henneberger et al. (2023). For our analysis, we include an Allsky camera (E9382-EHV, Vivotek, snapshots (see Fig. 1) and videos (please find video supplements here: Omanovic et al. (2024b))), a microwave radiometer (HATPRO G5, RPG) for continuous atmospheric temperature profiles, a ceilometer
(CHM 15k, Lufft) for the aerosol layer height detection, and cloud products from Cloudnet (Illingworth et al., 2007). The latter is a software based on python (Tukiainen et al., 2020) taking into account observations from cloud radars, lidar, microwave radiometer and model input from numerical weather prediction models to characterize clouds. This includes the determination of liquid water content (LWC) and ice water content (IWC), both of which we use to validate our model simulations, as well as the liquid and ice water path (LWP and IWP, respectively).

**2.1.2 CROSSINN**

The Cross-Valley Flow in the Inn Valley Investigated by Dual-Doppler Lidar Measurements (CROSSINN) campaign took place in summer and autumn of 2019 in the Inn Valley, Austria, with a focus on thermally-induced circulations and the mountain boundary layer. The Inn Valley is a major east-to-west oriented Alpine valley with a peak-to-peak distance of 10 km, while the valley bottom extends 5 km at our location of interest, located around 30 km east of the city of Innsbruck (Fig. 1a). An overview
over the campaign and the employed instruments can be found in Adler et al. (2021b). All instruments (Tab. 1) mentioned in

the next paragraph are located at the valley floor together with the so-called i-Box turbulence flux towers (Rotach et al., 2017). In our study focusing on clouds, we use observations from a ceilometer (CHM 15k, Lufft GmBH) for a proxy to cloud base, a microwave radiometer (HATPRO-G4, Raymetrics S. A.) for LWP observations, and radiosonde launches (every three hours) for vertical profiles of temperature and wind. All-sky images and animations are obtained from a visible and infrared camera (MX-S15D, Mobotix AG).

## 2.2 Model setup and simulations

We employed the numerical weather prediction model ICON (v2.6.6) (Zängl et al., 2015) in limited-area mode with varying horizontal resolutions (from 1 km down to 65 m, see Fig. 1a) and 80 vertical levels with the model top at 22 km (see Fig. A1). A recent study by Schmidt et al. (2024) showed that for varying vertical resolutions (on a global storm-resolving scale, i.e. 5 km) no convergence could be found for the microphysical properties. Also, tests conducted with the model setup (but for different case studies) showed no improvement of cloud representation or temperature inversions (Schöni, 2023) with increasing vertical resolution. Hence, we decided to only vary the horizontal resolution, and keep the vertical resolution the same as it is used in the operational model setup by the Swiss Federal Office of Meteorology (MeteoSwiss). While the 1 km domain covers the Alps, we have two smaller domains for our two separate measurement locations with $\Delta x = 65$ m; both are nested in a larger $\Delta x = 125$ m domain. Both the 1 km and 125 m domains receive their initial and boundary conditions (hourly update) from the COSMO-1 analysis (Schmidli et al., 2018) generated by MeteoSwiss. A brief comparison of cloud representation in the COSMO-1 analysis and the resulting ICON runs ($\Delta x$=1 km) revealed realistic cloud patterns in the COSMO analysis data and no large discrepancies with ICON. This setup is similar to Schemann et al. (2020) and Schemann and Ebell (2020), who found that constraining the model with a driving model (COSMO-1) yields an improved model performance compared observations of cloud properties. Furthermore, the impact of the driving model on the nested model should be minimal compared to the internal model variability and errors (Davies, 2014). The boundary conditions for the innermost domain (65 m) are updated every 30 min. The model time steps are 10 s, 1 s, and 0.5 s, respectively. We use high-resolution static input data for all domains, we employ the ASTER dataset with $\Delta x = 30$ m (NASA/METI/AIST/Japan Spacesystems and U.S./Japan ASTER Science Team, 2009) for topography, for land-use, the CORINE dataset (European Environmental Agency, 2017), and for soil properties, the Harmonized World Soil Database (FAO/IIASA/ISSCAS/JRC, 2012). Radiation is parameterized with the ecRAD scheme after Hogan and Bozzo (2018). Since our model set-up operates at $\Delta x = 1$ km and below, both the deep and shallow convection schemes are switched off. Mixing is achieved with a Smagorinsky-type (Lilly, 1962; Smagorinsky, 1963) turbulence scheme implemented by Dipankar et al. (2015), frequently used in ICON simulations in the kilometric and hectometric range (e.g., Heinze et al., 2017a; Hohenegger et al., 2023; Goger and Dipankar, 2024). The turbulence scheme is coupled to a surface-exchange scheme after Louis (1979) and the soil model TERRA_ML with 8 soil levels (Schulz and Vogel, 2020).

Given that we investigate the model performance in terms of cloud microphysics schemes, we conduct simulations with the one-moment microphysics (1M, Seifert, 2006; Doms et al., 2021) and the two-moment microphysics scheme (2M, Seifert and Beheng, 2006) for all resolutions. The former tracks only mass mixing ratios for cloud droplets, ice crystals, snow, rain, and graupel, while the latter also tracks the number mixing ratios for the same hydrometeors and in addition also hail. Often these

parameterizations are based on laboratory experiments or simplified theoretical concepts (Pruppacher and Klett, 1978). We provide here a short description of the processes relevant for our analysis and their representation in the bulk microphysics.

– **Cloud droplet activation:** The first and foremost process is cloud droplet activation, which requires the hygroscopic growth of aerosols, that act as cloud condensation nuclei (CCN). A cloud droplet is activated once it experiences spontaneous growth after passing a critical supersaturation with respect to water (generated through updrafts, i.e. adiabatic

cooling of air) and critical size (Lohmann et al., 2016). In 1M, the cloud droplet number concentration is prescribed given that only mass mixing ratios are predicted and "activation", i.e. the prescribed concentration, only occurs when supersaturated conditions are met (Doms et al., 2021). The prescribed cloud droplet number concentration is $200\,\mathrm{cm}^{-3}$. Hence, no hygroscopic growth of aerosol particles occurs. In the case of 2M, where both mass and number mixing ratios are prognostic, the cloud droplet number concentrations are calculated based on updraft (measure for supersaturation),

prescribed number of CCN ($250\,\mathrm{cm}^{-3}$), and their radius and a constant parameter account for hygroscopicity. For computational efficiency, this is achieved with so-called look-up tables that contain pre-calculated values for the activated cloud droplet number concentrations based on a matrix of the aforementioned parameters (Walko et al., 1995; Feingold et al., 1998; Seifert and Beheng, 2006). This cloud process already highlights the different levels of sophistication between the two bulk schemes.

– **Saturation adjustment:** We furthermore want to highlight a method that is introduced in the model to simplyfy the growth and evaporation of cloud droplets. Instead of following a mass-growth equation for cloud droplets (Lohmann et al., 2016), so-called saturation adjustments are performed before and after the cloud microphysics. In the saturation adjustment, the water vapor saturation pressure is calculated in the current grid box, and according to the water vapor deficit / surplus, cloud droplets will evaporate / grow (through condensation), respectively, to achieve saturation. This

only applies to the liquid phase, hence ice crystals can experience supersaturation with respect to ice.

– **Ice nucleation:** Another fundamental process is the formation of ice crystals in clouds, which can occur in mixed-phase clouds (both liquid and ice phase present) or ice clouds (i.e. cirrus clouds). In ice clouds, we find the process called homogeneous nucleation, which is the freezing of solution droplets in the atmosphere and only occurs at temperatures below -38 °C. For the temperature regime between -38 °C and 0 °C again aerosols (ice nucleating particles (INPs)) are

required to help form ice crystals (heterogeneous nucleation) (Lohmann et al., 2016). A broad range of aerosols can act as INPs (Kanji et al., 2017), the most prominent and abundant being dust particles. In models with 1M, the ice crystal number concentrations are prescribed as a function of temperature, while with 2M dedicated parameterizations for ice nucleation with prescribed aerosols are available. Most often a parameterization only for dust particles is implemented (e.g., Phillips et al., 2008; DeMott et al., 2015; Hande et al., 2015), which is a deterministic equation with a temperature

dependence and a freezing onset temperature. We decided to follow the ice nucleation scheme by Phillips et al. (2008) for dust particles, which can form ice at temperatures below -8 °C.

**Table 2.** Naming convention used in this study for the single studies. We differ between the two field campaigns by their topography (hilly vs. complex) and further distinguish between the horizontal resolutions of $\Delta x = 1$ km and $\Delta x = 65$ m and the employed microphysics scheme (one-moment (1M) and two-moment (2M) microphysics scheme).

| Field Campaign | CLOUDLAB (hilly terrain) | | | | CROSSINN (complex terrain) | | | |
|---|---|---|---|---|---|---|---|---|
| Resolution | 1 km | | 65 m | | 1 km | | 65 m | |
| Microphysics scheme | 1M | 2M | 1M | 2M | 1M | 2M | 1M | 2M |
| Simulation name | HT1 1M | HT1 2M | HT65 1M | HT65 2M | CT1 1M | CT1 2M | CT65 1M | CT65 2M |

– **Hydrometeor growth:** In both bulk schemes, the hydrometeors can collide and coalesce (cloud droplets), respectively aggregate (ice crystals) and thus form larger particles, such as raindrops and snow, and sediment (and possibly precipitate) if they reach larger sizes. All of which require additional parameterizations (Doms et al., 2021; Seifert and Beheng, 2006). Another process we want to highlight is the Wegener-Bergeron-Findeisen process (Wegener, 1911; Bergeron, 1935; Findeisen, 1938), which takes place when both the liquid and ice phase are present. It describes the growth of ice crystals through water vapor deposition, while cloud droplets evaporate due to the reduction in supersaturation. This way, cloud droplets act as a water vapor source for ice crystals to grow (Korolev and Mazin, 2003; Korolev, 2007). Through this process, mixed-phase clouds can be glaciated, i.e. the liquid phase evaporates and only the ice phase is left. It is known that the Wegener-Bergeron-Findeisen process is crucial for the cloud lifetime and its radiative effects, and weather and climate models struggle to accurately represent it (Liu et al., 2011; Kay et al., 2016; Klaus et al., 2016; McIlhattan et al., 2017; Kretzschmar et al., 2019; Huang et al., 2021; Omanovic et al., 2024a).

By combining the two field campaigns, we can conduct a model validation study in hilly (CLOUDLAB) and complex (CROSSINN) terrain. For clarity, we named the performed simulations after their topography, resolution, and the microphysics scheme, i.e. "HT1 1M" stands for a hilly terrain simulation with $\Delta x = 1$ km and the one-moment microphysics scheme, while "CT65 2M" describes a complex terrain simulation with $\Delta x = 65$ m and the two-moment microphysics scheme (see Table 2). For model validation, we use the so-called `meteogram` output as in Schemann et al. (2020), a single-point data output stream at the frequency of the model time step (10 s for $\Delta x = 1$ km and 0.5 s for $\Delta x = 65$ m, respectively). To ensure correct spatial representation, we take an spatial average of five meteograms with one being at the location of the field site, and the other four being equally distributed in a 100 m radius around the field site. For comparing the cloud characteristics, we apply a threshold value of 0.01 g m$^{-3}$ for liquid and ice water content to the observations and simulations to only consider in-cloud values.

## 2.3 Case study selection

The selection of the case studies was (1) limited by the operation times of the field campaigns, (2) the availability of observational data and (3) the occurrence of non-precipitating clouds. The latter was chosen because some observational instruments cannot measure reliably during precipitation events (e.g. microwave radiometer) or the reflectivity signal of remote sensing instruments is saturated (e.g. cloud radars). Hence, we decided to focus on the following cloud types: altocumulus (see Fig. 1f,

h, and i) and stratocumulus (see Fig. 1g) clouds. We identified the cloud types based on their height and the aerosol layer height, which serves as an indicator for the boundary layer height (see Fig. 3a and c, 7a and c). If the cloud was above the aerosol layer height, we classified it as an altocumulus cloud, otherwise as a stratocumulus cloud.

## 3 Results

### 3.1 Hilly terrain (HT)

In the following, we discuss the two case studies for hilly terrain (CLOUDLAB field site) first by comparing the cloud cover extent for HT1 and HT65 1M/2M, and then looking at the cloud characteristics, such as liquid and ice water content (LWC and IWC, respectively). While we do not have any observations of cloud droplet and ice crystal number concentrations, we include a discussion in the next section (respective figures for other cases in Appendix D) regarding the diagnosed (1M) and predicted (2M) number concentrations for all cases in hilly and complex terrain.

### 3.1.1 20 Mar 2022: Altocumulus clouds

The 20 Mar 2022 case study was characterized by weak westerly winds due to a weak low pressure system north-west of Switzerland (see Fig. 1b) which led to the formation of altocumulus clouds during nighttime before dissolving in the morning hours (see video supplements (Omanovic et al., 2024b) and Fig. 3a and c). We first compare the cloud cover extent for HT1 and HT65 in 1M and 2M configuration (Fig. 2) for the closest time step to the Allsky camera snapshot (see Fig. 1f). The cloud cover is diagnosed based on the present LWC and IWC in the middle of the simulated cloud and ranges from 0 to 100 %. Both resolutions simulate patchy clouds passing through the model domain, with HT1 showing larger and more coherent cloud structures, which is to be expected given the resolution. In general, HT65 predicts a larger fractional cloud cover and a longer-lived cloud (see video supplements (Omanovic et al., 2024b)). Looking at the normalized frequency distribution (Fig. B1), we do not see large differences between the two resolutions with respect to their cloud cover distribution. For the comparison between 1M and 2M, we see no clear signal for both resolutions but rather a spatial shift (HT1) or smaller extent of the cloud (HT65), with the largest differences at the cloud edges, which may be due to differences in turbulent mixing and also numerical diffusion. The similarity between the model setups is also notable in Fig. B1, where both configurations are fairly similar.

Next, we look at the liquid and ice water content and path (LWC/LWP and IWC/IWP, respectively) in the observations and the model simulations (Fig. 3). From the observations, we see that a cloud appears shortly after 3:00 UTC and thickens with time. The LWC occurs only sporadically with values of up to $15\,\mathrm{g\,m^{-3}}$, while the rest of the cloud is attributed by Cloudnet to be in the ice phase (Fig. 3c). Given the cold temperatures at cloud height (Fig. B2, $\sim$ -10 °C) ice crystals can form by heterogeneous nucleation on aerosols (Kanji et al., 2017) and lead to a full glaciation of the cloud due to favored growth of ice crystals in subzero temperatures compared to cloud droplets due to their difference in water vapor saturation pressure (i.e. the Wegener-Bergeron-Findeisen process).

**Table 3.** Mean and standard deviation of liquid and ice water content (g m$^{-3}$, LWC and IWC, respectively) averaged over the entire cloud and liquid and ice water path (g m$^{-2}$, LWP and IWP, respectively) for both resolutions and cloud microphysics schemes for the case on 20 Mar 2022 shown in Fig. 3. Only grid points with a LWC and IWC $> 0.01$ g m$^{-3}$ were included. Based on these values LWP and IWP were calculated.

|  | Observation | HT1 | | HT65 | |
|---|---|---|---|---|---|
|  |  | 1M | 2M | 1M | 2M |
| LWC | $2.49 \pm 3.24$ | $0.074 \pm 0.061$ | $0.159 \pm 0.146$ | $0.053 \pm 0.031$ | $0.037 \pm 0.022$ |
| LWP | $0.17 \pm 2.43$ | $0.188 \pm 0.293$ | $0.483 \pm 0.671$ | $0.172 \pm 0.138$ | $0.183 \pm 0.133$ |
| IWC | $0.07 \pm 0.07$ | $0.071 \pm 0.055$ | $0.037 \pm 0.014$ | $0.059 \pm 0.044$ | $0.099 \pm 0.084$ |
| IWP | $0.31 \pm 1.30$ | $0.482 \pm 0.595$ | $0.268 \pm 0.166$ | $0.345 \pm 0.356$ | $0.363 \pm 0.847$ |

When looking at the results from HT1 1M, we see that the model simulates a short-lived ice cloud (LWC/LWP are close to zero) around the time of the observed cloud (Fig. 3e). HT1 2M, however, simulates a liquid layer above 3000 m (Fig. 3i), which persists for several hours. The origin of this liquid layer may stem from the positive vertical velocities (i.e. updrafts) in
that region, that are more prominent in 2M than in 1M (see Fig. 4g). We see strong differences between HT1 and HT65. For the latter, 1M and 2M are able to simulate an ice cloud with a time delay of about one hour and two distinct clusters as observed by the remote sensing instruments (Fig. 3c, h and l), while 2M shows a higher IWC/IWP than 1M, which also impacts the vertical structure of temperature (Fig. B2h). This difference may come from slightly higher updrafts inside the cloud invigorating ice formation (see Fig. 4h). In both configurations, LWC is very low and occurs only sporadically (Fig. 3f and j). While there is
qualitative agreement between the model simulations and observations, we see a large discrepancy in the total amount of liquid exemplified by the LWP shown in Fig. 3b and by the mean and the standard deviation in LWC listed in Table 3. The model underestimates the LWC by a factor of 3, while the LWP agrees better. The LWP measurements shown in Table 3 are based on a microwave radiometer. The LWC is based on the Cloudnet algorithm, which combines several instruments and model data to classify the cloud. In this case, the classification as a liquid cloud occurs only sporadically, leading to a very high LWC
given the measured LWP. Hence, the interpretation of the LWC should be handled with care. For the ice phase, the agreement between the observations and simulations is better (see Table 3), except for HT 2M, which strongly underestimates the IWC and IWP.

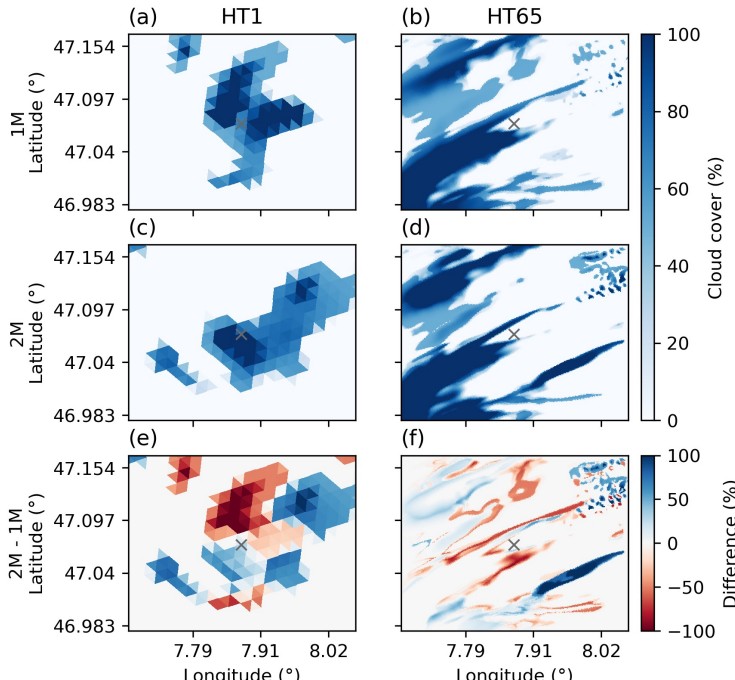

**Figure 2.** Comparison of fractional cloud cover (%) at ∼ 2'500 m AGL for the case study on 20 Mar 2022 at 05:50 UTC (closest time step to the allsky camera snapshot in Fig. 1f) for HT1 (**a**, **c**, **e**) and HT65 (**b**, **d**, **f**) in 1M (first row) and 2M (second row) configuration, and their difference (2M - 1M, last row). "x" marks the CLOUDLAB field site. The domain of HT1 was zoomed in to show the same extent as the HT65 does.

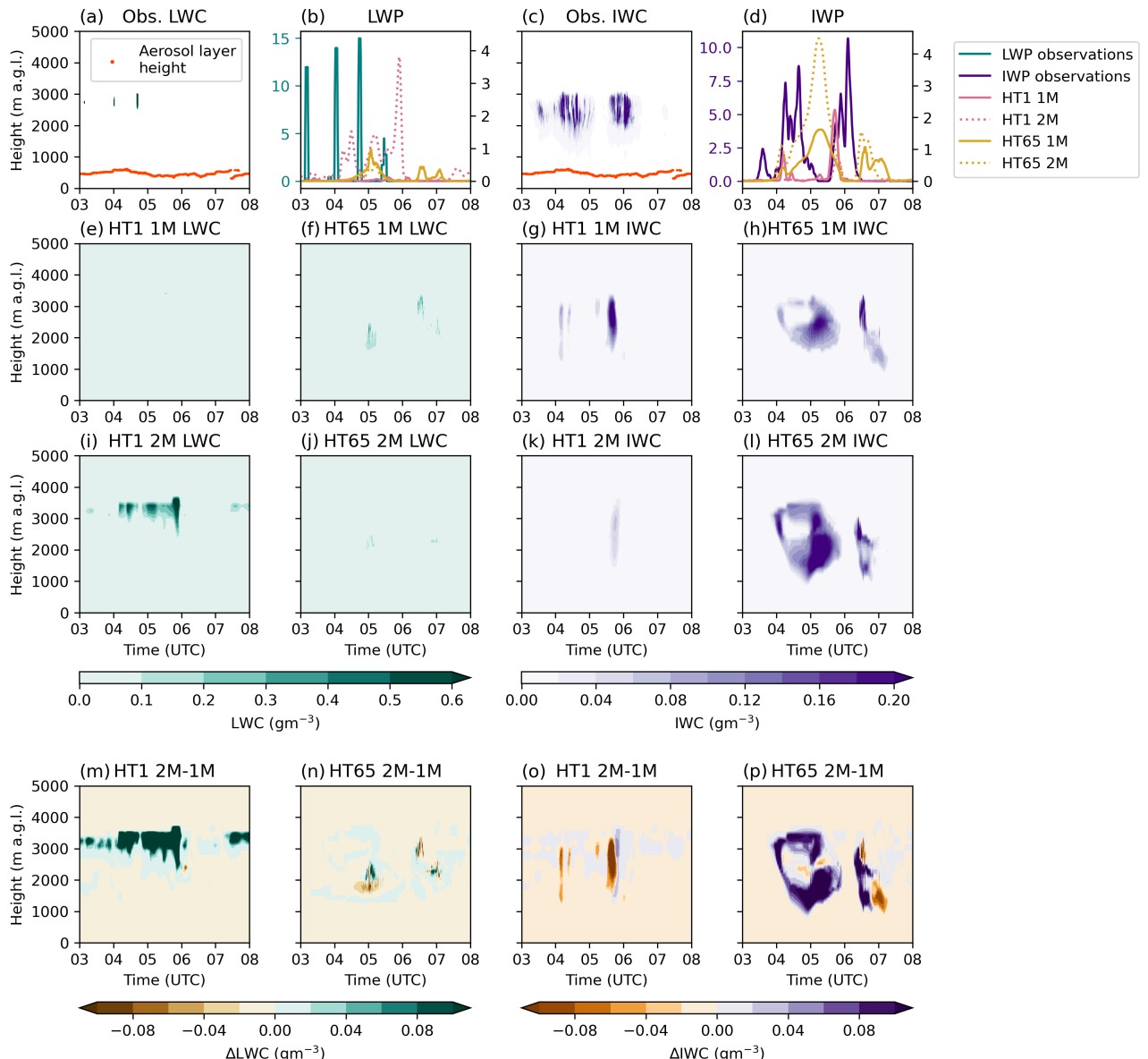

**Figure 3.** Overview of cloud characteristics for case study 20 Mar 2022. **(a)** and **(c)** show the liquid water content (LWC, $g\,m^{-3}$) and ice water content (IWC, $g\,m^{-3}$) based on the algorithm by Cloudnet with the aerosol layer height (orange dots) serving as a proxy for the boundary layer height. **(e)**-**(l)** show the model responses for both LWC and IWC for both resolutions (HT1 and HT65) and both bulk microphysics schemes (1M and 2M). **(b)** and **(d)** show the observed (left y-axis) and the simulated (right y-axis) LWP and IWP ($g\,m^{-2}$), respectively.

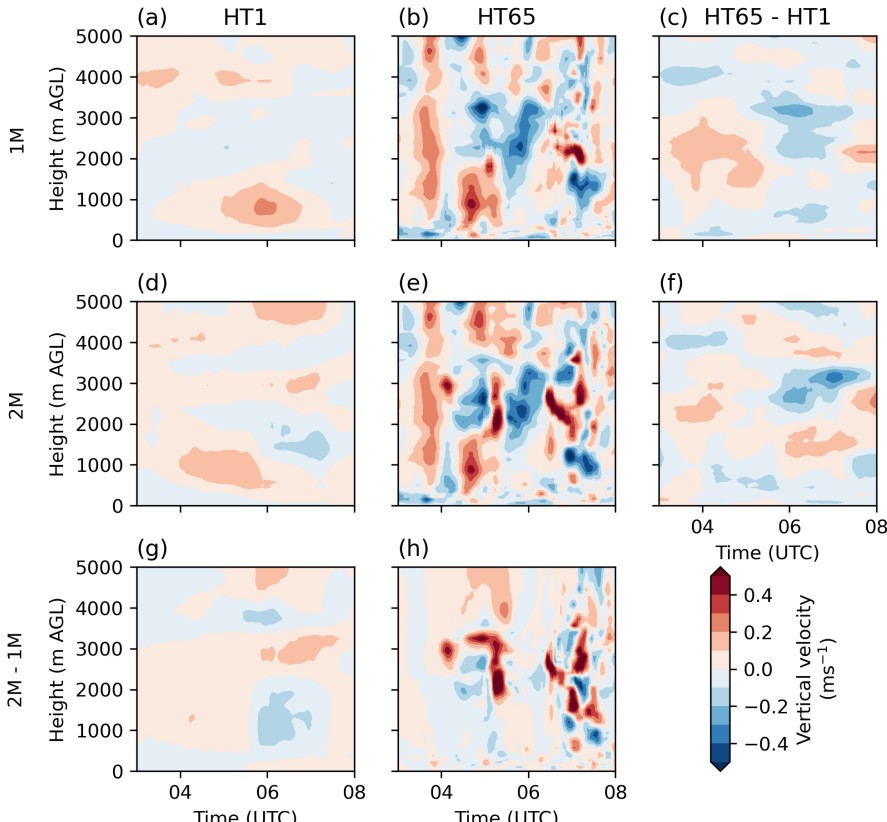

**Figure 4.** Hovmöller diagrams for vertical velocity ($\mathrm{m\,s^{-1}}$) averaged over $10\,\mathrm{min}$ for HT1 and HT65 with 1M and 2M, respectively, for 20 Mar 2022. The last column shows the differences between the resolutions (HT65 - HT1), where HT65 was averaged over a running mean of $10\,\mathrm{s}$ to have the same time frequency as HT1. The last row shows the differences between 2M - 1M for both HT1 and HT65.

Figure 5 (for the second case study: see Appendix D, Fig. D1, for the simulations in complex terrain see Fig. D2) shows the temporal evolution of the cloud droplet and ice crystal number concentrations (CDNC and ICNC, respectively) for both resolutions and both microphysics schemes for the hilly terrain case studies. As mentioned, the CDNC are prescribed in 1M (=$200\,\mathrm{cm^{-3}}$) as soon as a cloud forms. In the model CDNC is only used for calculating the collection kernels between the droplets, and it does not change over time. In the figures, the markers are only set for illustrative purposes highlighting when the LWC exceeded $0.01\,\mathrm{g.m^{-3}}$ in 1M simulations. In 2M, CDNC are predicted, and thus change with time and depend on the cloud droplet activation and removal due to collision processes (see Sect. 2). In both schemes, the collision and coalescence rate of cloud droplets (i.e., autoconversion) may be low. In 1M the high CDNC and low LWC lead to small cloud droplets limiting collisions and in 2M low CDNC and low LWC also yield small autoconversion rates. The two microphysics schemes differ strongly in CDNC, with the prescribed concentration in 1M probably being a better estimate for the CDNC for a continental cloud, while 2M predicts rather low concentrations. This could be a consequence of interactions with the ice phase, where at subzero temperatures, the ice phase is the favored state, and thus ice crystals will form and grow at the expense of evaporating

cloud droplets. One hypothesis is, that this balance in the model is more on the side of the ice crystals. This is further supported by the strong underestimation of the LWC/LWP in the simulations. For ICNC we see that 1M strongly underestimates it, which may arise from the equation for ICNC from Cooper (1986), where at temperatures around -10 °C the ICNC activity is underestimated. For 2M we see that only for the HT65 simulation a realistic ICNC is simulated with concentrations maximizing at 0.1 cm$^{-3}$, while for HT1 ICNC is by almost three magnitudes of order too low, also almost no IWC/IWP was simulated.

Hence, while the CDNC is strongly underestimated in 2M simulations, which may come from the balance between the liquid and ice phase. This is an issue weather and climate models struggle with (Liu et al., 2011; Kay et al., 2016; Klaus et al., 2016; McIlhattan et al., 2017; Kretzschmar et al., 2019; Huang et al., 2021; Omanovic et al., 2024a), we see a more realistic simulation of ICNC than for 1M. For the complex terrain case studies in summer, we only investigate CDNC (Fig. D2). The concentrations are slightly higher (factor 2) than for the HT simulations, but this is probably still an underestimation as higher

CDNC over land can be expected (Lohmann et al., 2016).

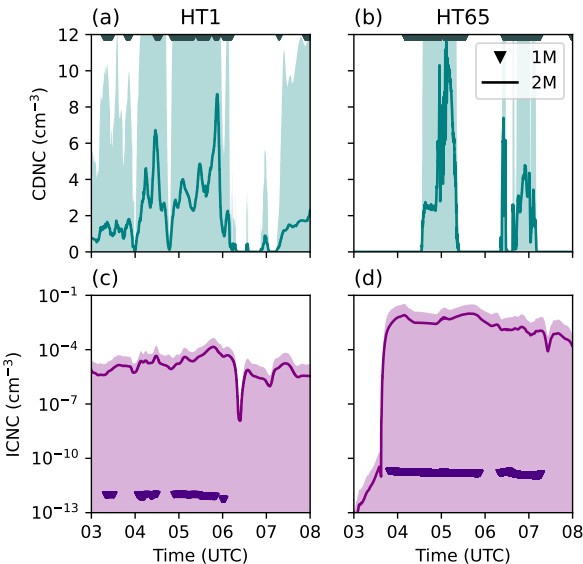

**Figure 5.** Simulated cloud droplet (**a** and **b**) and ice crystal (**c** and **d**) number concentrations (CDNC and ICNC, respectively) for 20 Mar 2022 for both resolutions (HT1 and HT65, **a/c** and **b/d**, respectively). The number concentration from 1M are shown as markers, whenever there is a cloud present. In 1M CDNC = 200 cm$^{-3}$ (prescribed, markers only for indicating cloudy conditions), while ICNC is calculated as a function of temperature following Cooper (1986) (diagnosed). The predicted quantities from 2M are shown as means (solid lines) ± standard deviations (shadings).

### 3.1.2  31 Jan 2023: Stratocumulus clouds

The second case we investigate are stratocumulus clouds that formed in the afternoon of 31 Jan 2023. The weather was characterized by north-westerly winds (see Fig. 1c). The cloud lived for several hours and at times ice crystals sedimented towards the ground causing a large vertical extent of the cloud (see Fig. 7a and c).

**Table 4.** Mean and standard deviation of liquid and ice water content in g m$^{-3}$ (LWC and IWC, respectively) avergared over the entire cloud and liquid and ice water path in g m$^{-2}$ (LWP and IWP, respectively) for both resolutions and cloud microphysics schemes for the case on 31 Jan 2023 shown in Fig. 7. Only grid points with a LWC and IWC $> 0.01$ g m$^{-3}$ were included. Based on these values LWP and IWP were calculated.

| | Observation | HT1 | | HT65 | |
|---|---|---|---|---|---|
| | | 1M | 2M | 1M | 2M |
| LWC | $0.13 \pm 0.19$ | $0.145 \pm 0.125$ | $0.152 \pm 0.141$ | $0.023 \pm 0.016$ | $0.015 \pm 0.003$ |
| LWP | $0.55 \pm 0.88$ | $0.171 \pm 0.189$ | $0.446 \pm 0.454$ | $0.033 \pm 0.022$ | $0.018 \pm 0.004$ |
| IWC | $0.25 \pm 0.43$ | $0.035 \pm 0.025$ | $0.005 \pm 0.001$ | $0.036 \pm 0.042$ | $0.068 \pm 0.057$ |
| IWP | $1.46 \pm 7.13$ | $0.079 \pm 0.102$ | $0.005 \pm 0.012$ | $0.059 \pm 0.143$ | $0.156 \pm 0.327$ |

We again first compare the cloud covers at $2500$ m for HT1 and HT65 in Fig. 6, which shows the closest time step to the allsky camera snapshot in Fig. 1g. While we see a full cloud layer in the snapshot, both model resolutions simulate a more patchy cloud with HT65 showing clouds in the form of streaks and HT1 showing more grouped together clouds (see video supplements (Omanovic et al., 2024b)). Both resolutions cannot reproduce the observed cloud conditions. In terms of the performance of the cloud microphysics scheme, we see a very similar behavior as for the previous case study (20 Mar 2022), where the strongest differences between 1M and 2M occur at the cloud edges (due to differences in turbulent mixing and / or numerical diffusion), which appears to be more extreme in the case of HT1 given the coarser resolution. When looking at the normalized frequency distribution, we cannot distinguish any large differences (Fig. B3). This points to a more stochastic nature of the schemes, than to systematic changes caused by the choice of scheme.

Figure 7 shows the LWC/LWP and IWC/IWP for the observations and model simulations. We see a longer-lived cloud than in the previous case with a liquid layer that descends over time from $3000$ m to $2000$ m (Fig. 7a). We also see that the cloud produced very light precipitation (long, purple streaks reaching the ground and increase in the aerosol layer height in Fig. 7c). Both, HT1 1M and 2M, simulate a weaker liquid layer, whereas the liquid layer increases in height with time before it dissolves too early (Fig. 7e and i) and in the early night a liquid layer forms again. 1M and 2M have similar LWPs during the lifetime of the cloud, with 2M showing slightly higher values for the early afternoon hours. Similar to the other case study, LWP is underestimated in the model by a factor 2 (Fig. 7b). While Cloudnet classified parts of the cloud to be ice, the model barely simulates any ice (Fig. 7g and k), further highlighted by the very low IWP in Fig. 7d. We see a different picture for the case of HT65 (Fig. 7f, j, h, and l). We barely have any LWC present in neither 1M or 2M (Fig. 7f and j and Table 4). Both configurations simulate some ice clouds in the early afternoon, where 2M again shows a higher IWC/IWP than 1M. However, for both schemes the cloud is shorter-lived than in the observations failing to reproduce the longevity of the observed cloud. Including the vertical velocity in our analysis (Fig. 8), we see that HT65 in general has stronger vertical velocities than HT1 ($\pm 0.3$ m s$^{-1}$). This may lead to an invigorated ice formation as seen in both configurations, 1M and 2M.

31 Jan 2023 15:30

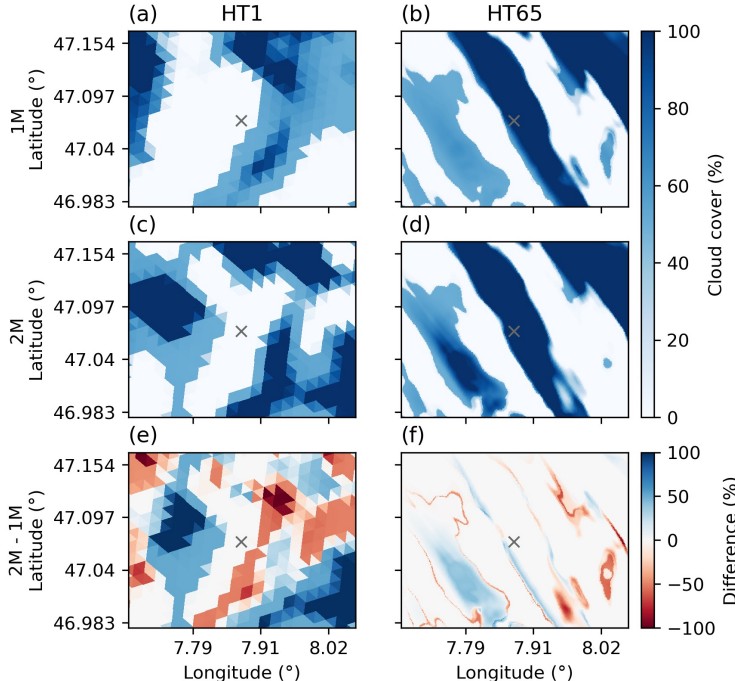

**Figure 6.** Comparison of fractional cloud cover (%) at ∼ 2'500 m AGL for the case study on 31 Jan 2031 at 15:30 UTC (closest time step to allsky camera snapshot in Fig. 1g) for HT1 (**a**, **c**, **e**) and HT65 (**b**, **d**, **f**) in 1M (first row) and 2M (second row) configuration, and their difference (2M - 1M, last row). "x" marks the CLOUDLAB field site. The domain of HT1 was zoomed in to show the same area as the HT65 does.

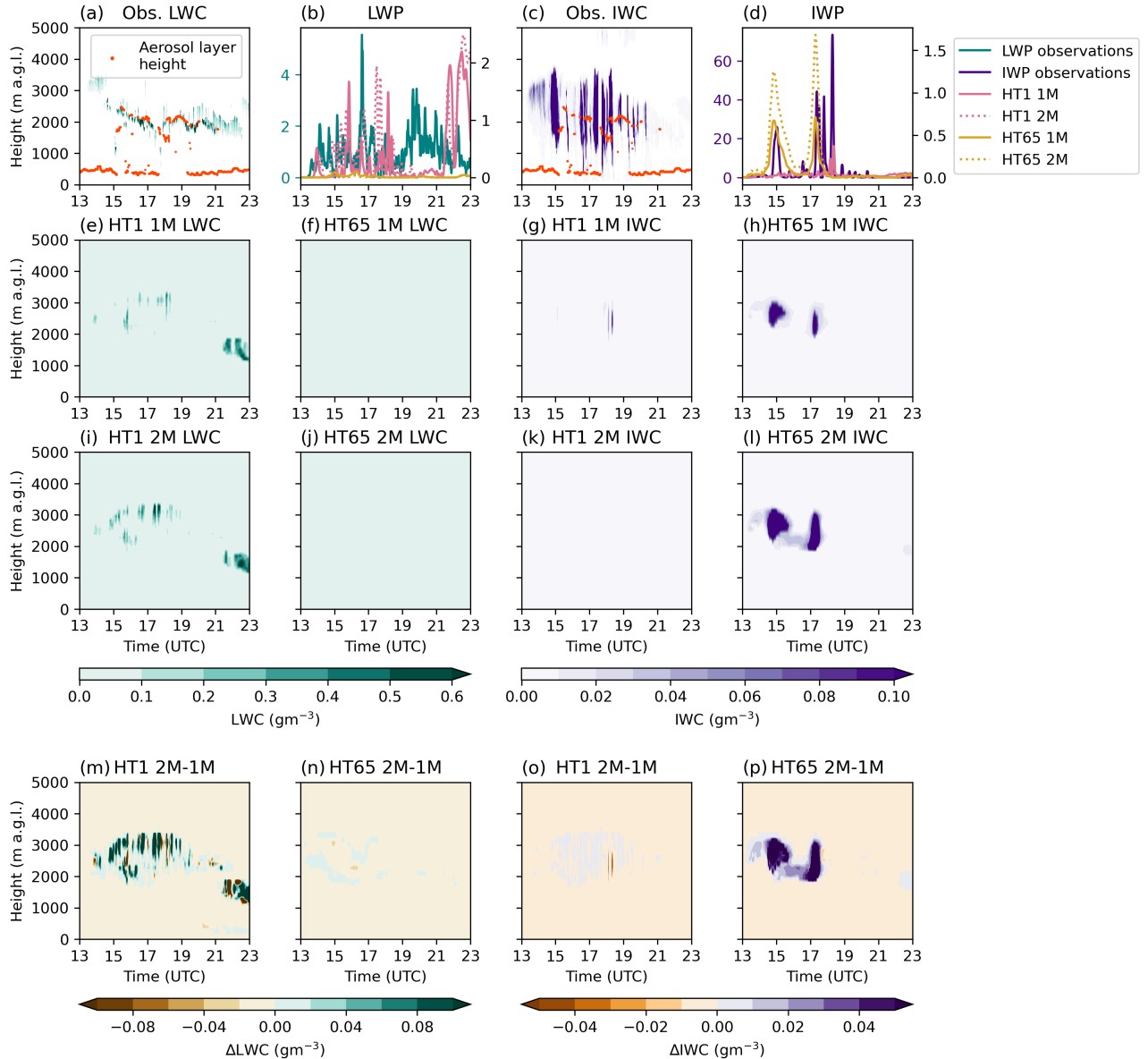

**Figure 7.** Overview of cloud characteristics for case study 31 Jan 2023. **(a)** and **(c)** show the liquid water content (LWC, $g\,m^{-3}$) and ice water content (IWC, $g\,m^{-3}$) based on the algorithm by Cloudnet with the aerosol layer height (orange dots) serving as a proxy for the boundary layer height. **(e)**-**(l)** show the model responses for both LWC and IWC for both resolutions (HT1 and HT65) and both bulk microphysics schemes (1M and 2M). **(b)** and **(d)** show the observed (left y-axis) and the simulated (right y-axis) LWP and IWP ($g\,m^{-2}$), respectively.

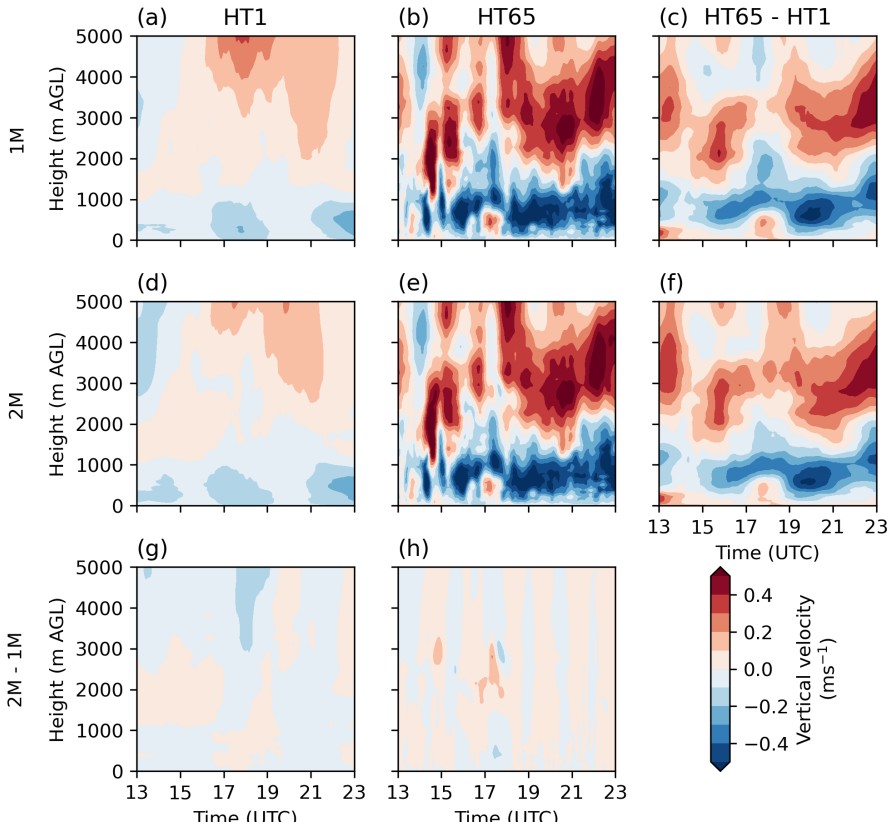

**Figure 8.** Hovmöller diagrams for vertical velocity ($\mathrm{ms}^{-1}$) averaged over $10\,\mathrm{min}$ for HT1 and HT65 with 1M and 2M, respectively, for 31 Jan 2023. The last column shows the differences between the resolutions (HT65 - HT1), where HT65 was averaged over a running mean of $10\,\mathrm{s}$ to have the same time frequency as HT1. The last row shows the differences between 2M - 1M for both HT1 and HT65.

## 3.2 Complex terrain (CT)

As mentioned in the description of the observations, the CROSSINN campaign's focus was not on cloud observations. Nevertheless, vertical profiles from radiosondes, backscatter from a ceilometer, an all-sky camera, and LWP observations from the HATPRO radiometer can be utilized for our model validation study. Since we are validating summertime case studies, the clouds in the CT simulations mostly consist of water and only liquid water content (LWC) is compared. Observations and simulations of vertical profiles of temperature (Fig. C1) suggest that the zero degree line is above $3000\,\mathrm{m\,a.\,g.}$ (Aug 5, 2019) and $2000\,\mathrm{m\,a.\,g.}$ (Aug 14, 2019), respectively. Given that ice clouds form at temperatures below -8 °C (Phillips et al., 2008), we assume that the clouds in the two following case studies do only consist of liquid water and we hereby only analyse the liquid water content.

### 3.2.1    05 Aug 2019: Altocumulus clouds

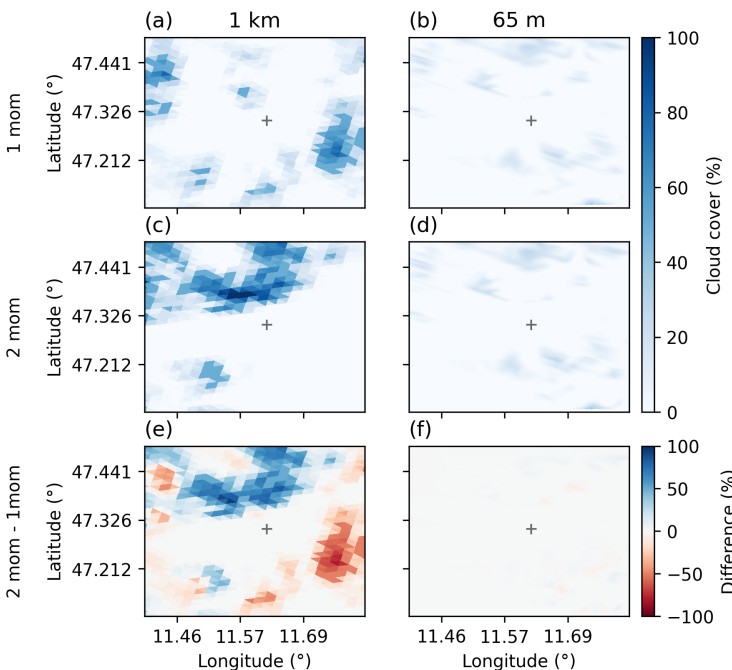

**Figure 9.** Comparison of cloud cover extent (%) at ∼ 4'000 m AGL for the case study on 05 August, 2019 at 09:00 UTC (closest time step to Allsky camera snapshot in Fig. 1h) for CT1 (**a**, **c**, **e**) and HT65 (**b**, **d**, **f**) in 1M (first row) and 2M (second row) configuration, and their difference (2M - 1M, last row). "+" marks the CROSSINN field site. The domain of CT1 was zoomed in to show the same area as the HT65 does.

August 5, 2019 was characterized by a high pressure system over central Europe (Fig. 1d), suggesting weak synoptic influence over the Alps. The Inn Valley was mostly dominated by low wind speeds below $4\,\mathrm{ms}^{-1}$, with down-valley flows during the night-time and up-valley flows during the daytime. The all-sky camera shot at 09:00 UTC in the Inn Valley suggests an altocumulus cloud layer covering the entire sky view, while the video animation (Omanovic et al., 2024b) suggests that this cloud layer persists, with occasional interruptions, for most of the day. The closest model time step to the all-sky camera snapshot shows large differences between the simulations and the observations (Fig. 9): At ≈4000 m above ground, the kilometric CT1 simulations show a patchy cloud cover over our area of interest. The cloud cover structure does not follow the underlying topography, mostly because these high clouds are already located above the highest peaks in the surroundings (above ≈3500 m a.m.s.l.). Interestingly, due to the patchiness of the cloud cover, there are no simulated clouds above our area of interest, the valley floor. The 2M scheme simulates a thicker cloud cover than the 1M run. The CT65 simulations suggest a more continuous cloud layer at ≈4000 m a.g., covering most of our area of interest. We note here that this "patchiness" will also affect our interpretation of the model results in the next paragraphs. As in the CT1 simulations, there is no clear signal whether the 1M scheme or the 2M scheme simulate "more" or "less" clouds (Fig. C2). The ceilometer observations from the

valley floor (Fig. 10a) show in the first hours a cloud-free night (00:00-02:00 UTC). After 03:00 UTC, a cloud layer with a cloud base height of around 400 m a.g. develops, and remains persistent until 10:00 UTC (see video supplements (Omanovic et al., 2024b)). In the CT65 simulations (Fig. 10b,c), the simulated LWC suggests that there are no clouds present in the first hours of simulations. Cloud development is delayed in the CT65 runs and high clouds only form after 10:00 UTC. However, after 10:00 UTC, the CT65 simulates a realistic cloud structure with a similar cloud base height of 4000 m a.g., and the cloud remains persistent until the end of our time of interest (18:00 UTC). Changing the microphysics scheme has indeed an impact on simulated LWC: The 2M scheme simulated higher LWC amounts compared to the 1M run, but still, only after 10:00 UTC (Fig. 10d). This behaviour is also visible in the simulated LWP (Fig. 10j), where the 2M scheme generally simulates higher LWP values of up to $12\,\mathrm{g\,m^{-2}}$. Unfortunately, a HATPRO failure occurred between 12:00 UTC and 17:00 UTC, so we can not validate the simulations during this time period.

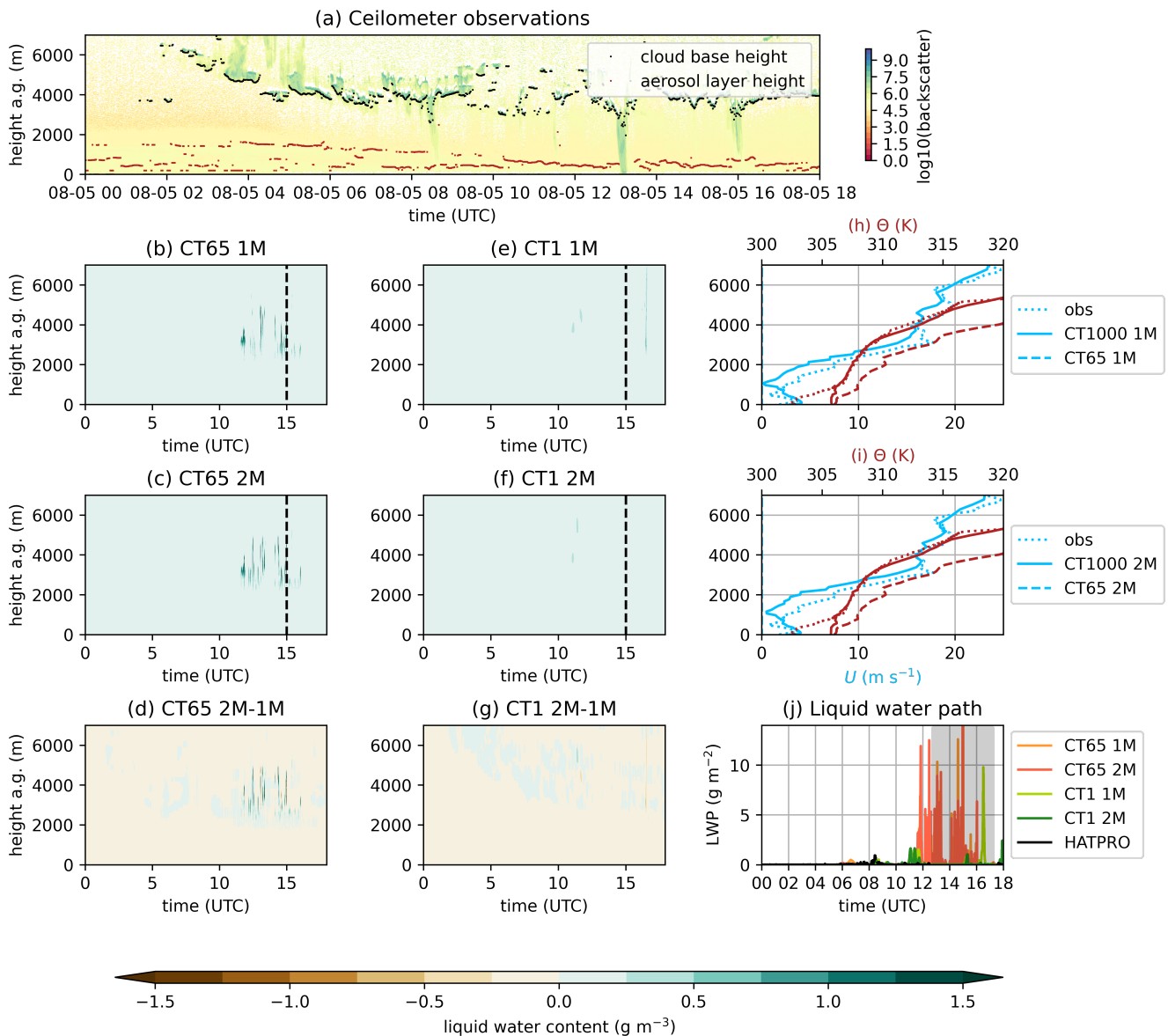

**Figure 10.** Aug 05, 2019, Inn Valley floor: **(a)** Ceilometer observations showing cloud backscatter (colors) with cloud base height (black dots) and aerosol layer height (red dots) determined by the instrument. Panels **(b)**-**(g)** Time series of model output of liquid water content from simulations CT65 1M (**b**, **c**) and CT1 1M (**e**, **f**) with 1M (**b**, **e**) and 2M (**c**, **f**), respectively. Panels **(d)** and **(g)** show the difference between the 2M and 1M schemes. The colorbar is valid for all panels showing LWC. **(h)**-**(i)** Vertical profiles along the dashed line in panels **b**, **e**,**c**,**f** of potential temperature (red) and horizontal wind speed (blue) of radiosonde observations (dots), CT1 (full lines), and CT65 (dashed lines) from 15:00 UTC. **(j)** LWP observations from HATPRO (black, instrument failure from 12:30 UTC–17:00 UTC, grey shaded area), and LWP model output from the respective simulations (colors).

The CT1 simulations (Fig. 10e,f), however, show almost no clouds compared to the CT65 simulations. There are no relevant LWC amounts simulated for most of the time with the only exception for very short-lived (dissipation after several minutes of simulation time) clouds with LWC values below $0.04\,\mathrm{g\,m^{-3}}$ at around 12:00 UTC. The choice of the microphysics scheme has no positive impact on the simulation of clouds in the CT1 runs (Fig. 10g) at the valley floor. The LWP shows only small values before 12:00 UTC and at around 16:00 UTC, but remains generally smaller than in the CT65 runs. A reason for the (almost) completely absent cloud formation can be found in the vertical velocity time series (Fig. 13): While the CT1 runs (Fig. 13f,j) are unable to resolve vertical motions, the CT65 runs (Fig. 13e,i) simulate continuous up- and downdrafts after 10:00 UTC, favoring cloud formation during this time period. The vertical profiles of observed and simulated potential temperature profiles at 15:00 UTC show a large discrepancy between model and observations at 2000 m a.g. (Fig. 10h,i): Both simulations, CT65 and CT1, underestimate the horizontal wind speed while simulating realistic potential temperature profiles. There is no clear indicator why this happens, but 2000 m a.g. is approximately the crest height of the surrounding mountains - and it is possible that the model is unable to simulate the local flow structure at crest height accordingly, leading to unrealistic circulations affecting cloud formation. Still, it has to be pointed out that the CT65 simulation is able to produce a cloud cover over the valley - because of basic small-scale flow features that are successfully simulated at sub-hectometric resolutions (Fig. 13e,f). Interestingly, as the video animation (Omanovic et al., 2024b) suggests, the CT1 simulation is able to produce altocumulus clouds, but mostly over the surrounding ridges, but not over the valley itself. This is also evident in the histograms (Fig. C2), suggesting that both mesh sizes produce similar amounts of cloud cover, independent of the microphysics scheme.

### 3.2.2 14 Aug 2019: Altocumulus clouds

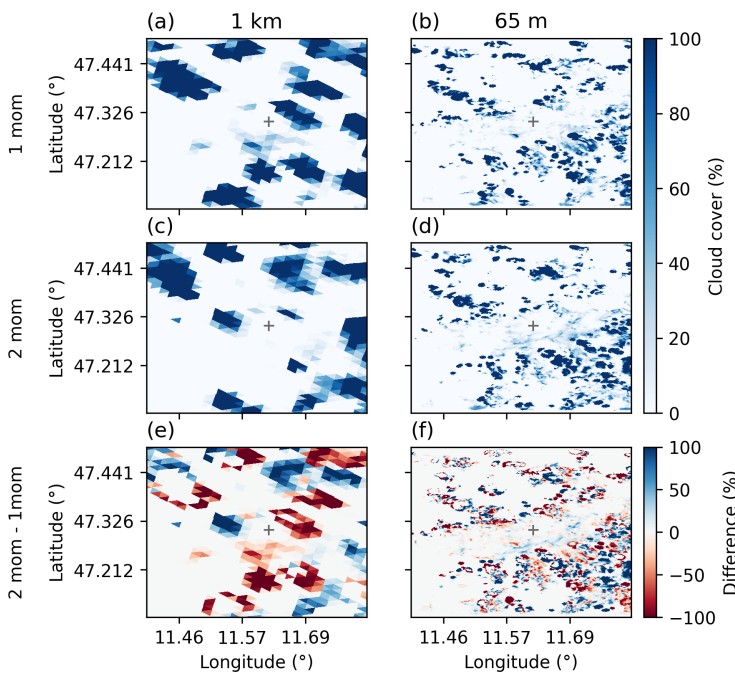

**Figure 11.** Comparison of cloud cover extent (%) at ∼ 4'000 m a.g. for the case study on 14 August, 2019 at 09:00 UTC (closest time step to Allsky camera snapshot in Fig. 1h) for CT1 (**a**, **c**, **e**) and CT65 (**b**, **d**, **f**) in 1M (first row) and 2M (second row) configuration, and their difference (2M - 1M, last row). "+" marks the CROSSINN field site. The domain of HT1 was zoomed in to show the same area as the HT65 does.

On August 14, 2019, the Alps were under the influence of westerly winds (see Fig. 1e). The local boundary layer in the Inn Valley was dominated by the formation of a thermally-induced valley wind circulation (Lehner et al., 2019), resulting in down-valley flows during the night-time and a distinct up-valley flow during the daytime. For this case study, we will mostly focus on the break-up of the low-lying nighttime stratus clouds and the following shallow-cumulus cloud formation. In the morning, a thick stratus layer with occasional precipitation was present in the valley. The stratus layer dissolved at around 06:00 UTC and throughout the rest of the day, mostly shallow altocumulus clouds were present over the valley and the surrounding mountains (see video supplements (Omanovic et al., 2024b)). A comparison with the simulations at 09:00 UTC shows that in the CT1 simulations the valley itself is cloud-free, but over the mountains, clouds are visible in both CT1 and CT65. In the CT1 runs, the 1M scheme simulates more clouds than the 2M scheme, although it is questionable whether the horizontal extent of the cumulus clouds over the mountains is realistic. In contrast, the CT65 simulations, show at the same time step small-scale altocumulus clouds scattered over the domain, especially over the mountain slopes, and this is in better agreement with the all-sky camera observations (Fig. 1i). Both microphysics schemes simulate the scattered cumulus clouds and major differences can only be seen in their location, but not in their pattern.

The ceilometer observations at the valley floor (Fig. 12a) suggest a low-lying cloud layer during the nighttime, likely stratus or nimbostratus clouds with occasional precipitation at 01:00 UTC and 03:00 UTC. However, our time period of interest starts only around 05:00 UTC after sunrise, when the stratus cloud layer (at ≈2000m a.g.) weakens and transforms to scattered altocumulus clouds (at ≈3000m a.g.), especially visible in the intermittent backscatter of the ceilometer after 06:00 UTC.

According to the observations, the cloud base height rises from around 2000 m a.g., together with a developing aerosol layer height, up to 3000 m a.g. during the daytime. A comparison with radiosonde observations and wind speeds suggests the development of a convective boundary layer, and the convection and up-slope flows lead to the formation of cumulus clouds over the surrounding mountain slopes and peaks and not over the valley floor itself. Still, the cumulus clouds are advected over the valley floor with the general up-valley flow, visible in the all-sky camera, leading to a transient cloud cover until around

14:00 UTC. After that, no clouds are recognizable in the ceilometer observations.

The CT65 simulation (Fig. 12b,c) suggests the presence of a low stratus layer with a cloud base height of around 2000 m before 06:00 UTC, in agreement with the ceilometer observations. After the breakup of the low stratus cloud layer at ≈ 06:00 UTC, the model simulates single smaller cumulus clouds with a cloud base height of around 3000 m a.g. over the valley floor, but compared to the ceilometer time series, the simulated cumulus clouds are fewer and weaker as well. Compar-

370 ing the two microphysics schemes (Fig. 12d), the 2M scheme generally leads to larger LWC values ($+1.2\,\mathrm{g\,m^{-3}}$) in the 65M runs during the time of the low stratus. There is no clear signal for the daytime cumulus clouds, because both microphysics produce occasional clouds - this is likely related to the very patchy (and transient) cloud cover also visible in Fig. 11. The simulated LWP (Fig. 12j) reveals that the CT65 runs simulate realistic values in agreement with the observations. However, there is a time shift, where the second maximum is simulated earlier (around 04:00 UTC), while the observations suggest the

375 second maximum to appear later (06:00 UTC). The CT1 simulations suggest a low-lying cloud layer before 06:00 UTC as well (Fig. 12e,f), but the simulated cloud base layer lies clearly above 2000 m, suggesting a stratus layer with smaller vertical extent than in the CT65 runs. Furthermore, the stratus layer in the CT1 runs is less persistent than in the CT65 runs - clearly visible in an interruption at around 04:00 UTC. However, the breakup of the stratus layer together with the evolution of the daytime boundary layer is simulated accurately, and in the afternoon, small-scale altocumulus clouds are present as in the observations.

As in the CT65 runs, the 2M leads to higher LWC contents (Fig. 12g). The CT1 runs simulate the LWP time series in a realistic way (i.e., LWP maxima in observations and model are synchronous), however, the absolute values of LWP are underestimated by the model by about $5\,\mathrm{g\,m^{-2}}$.

In contrast to the other case study in complex terrain (05 Aug 2019), the model is able to simulate realistic potential temperature and horizontal wind speed profiles in the valley for both resolutions (Fig. 12h,i), and this also leads to less discrepancies

in cloud formation. Furthermore, since local circulations are well-represented (Fig. 13g,k,h,l), there is no large difference in cloud cover between the two mesh sizes and microphysics schemes (Fig. 11).

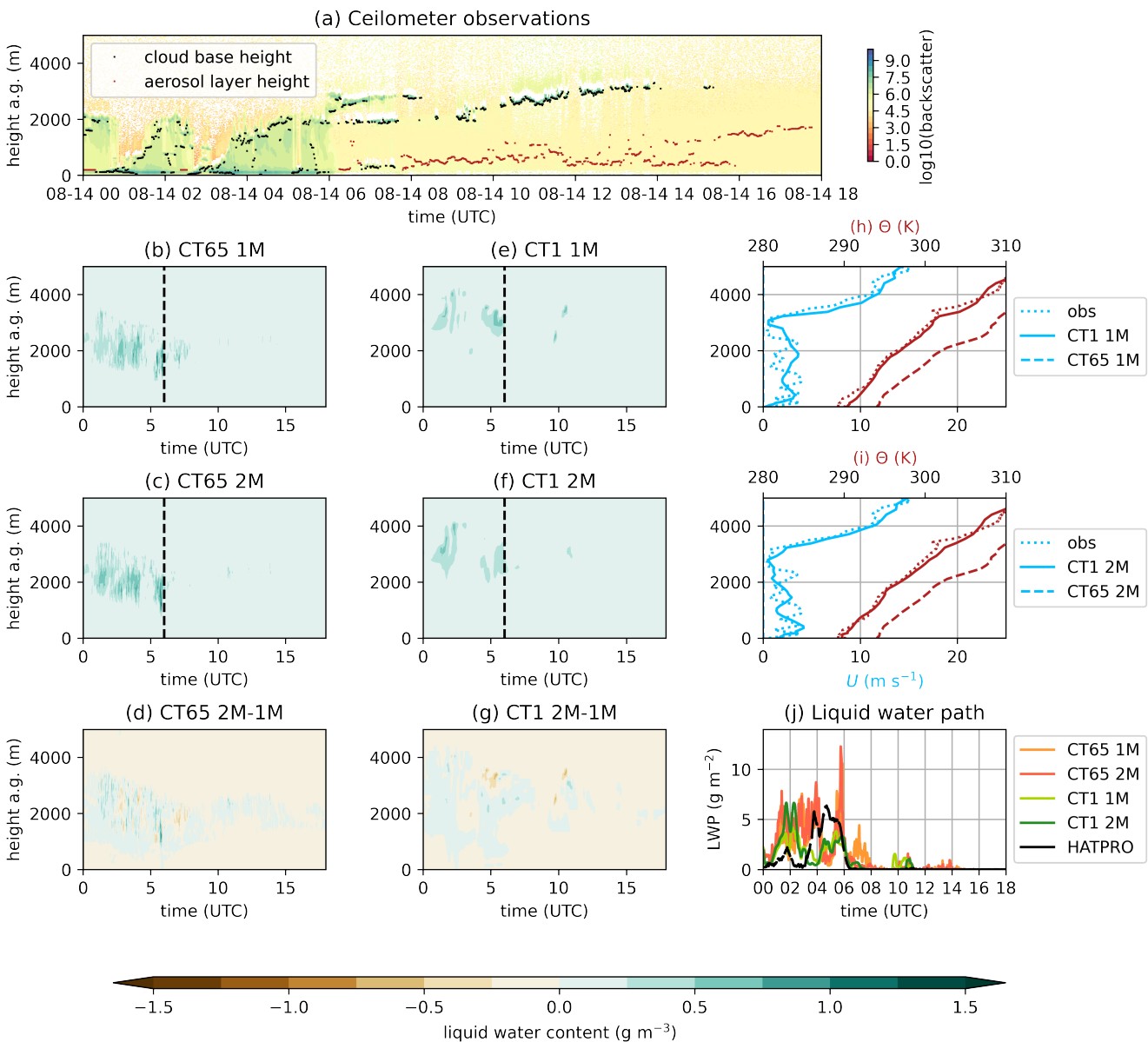

**Figure 12.** Aug 14, 2019, Inn Valley floor: **(a)** Ceilometer observations showing cloud backscatter (colors) with cloud base height (black dots) and aerosol layer height (red dots) determined by the instrument. Panels **(b)**-**(g)** Time series of model output of liquid water content from simulations CT65 1M (**b**, **c**) and CT1 1M (**e**, **f**) with 1M (**b**, **e**) and 2M (**c**, **f**), respectively. Panels **(d)** and **(g)** show the difference between the 2M and 1M schemes. The colorbar is valid for all panels showing LWC. Panels h-i) Vertical profiles along the dashed line in panels **b**, **e**,**c**,**f** of potential temperature (red) and horizontal wind speed (blue) of radiosonde observations (dots), CT1 (full lines), and CT65 (dashed lines) from 15:00 UTC. j) LWP observations from HATPRO (black), and LWP model output from the respective simulations (colors).

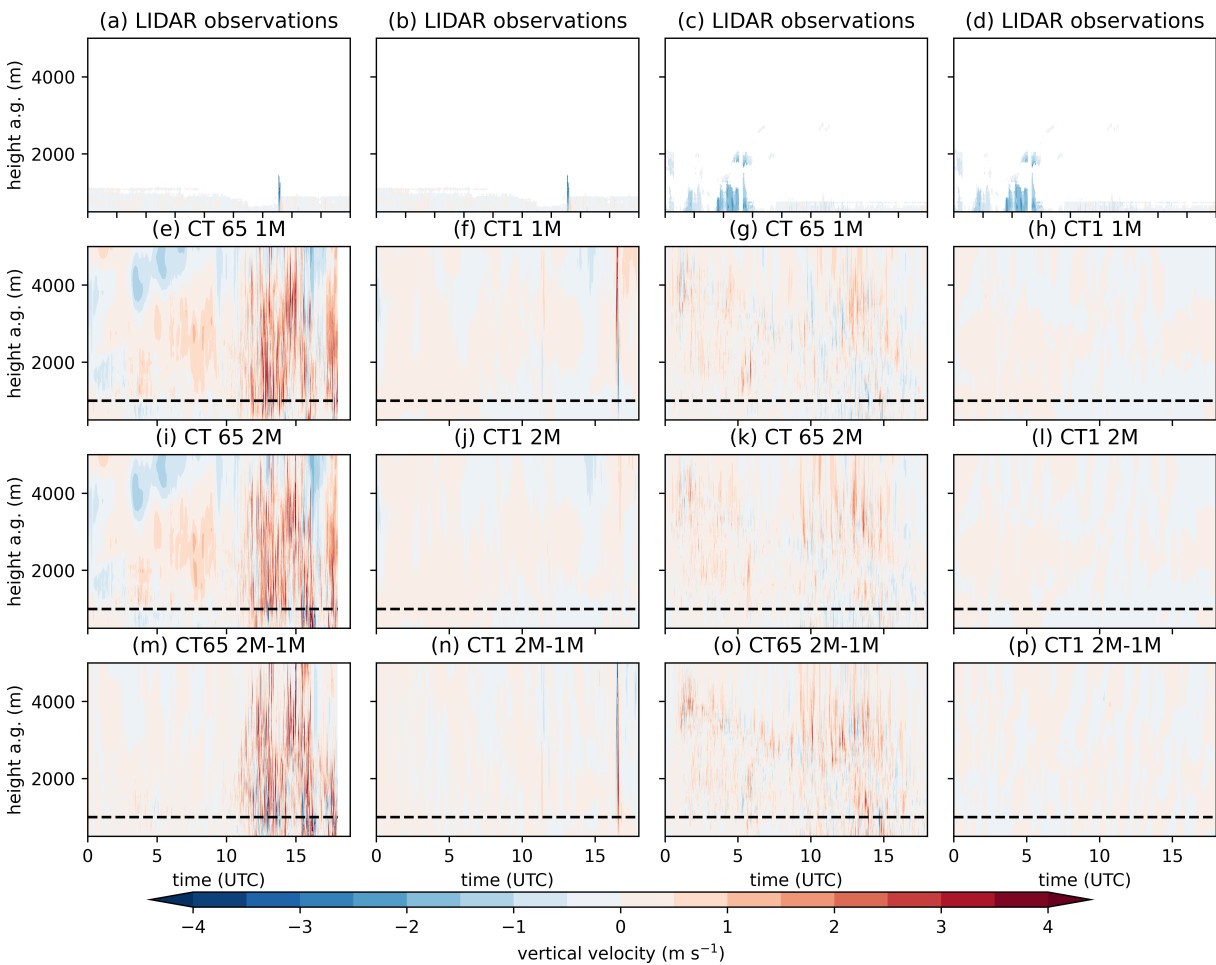

**Figure 13.** Time evolution of vertical profiles of vertical velocity from the SL88 LIDAR observations (panels **(a)**-**(d)**), and model output from the different horizontal resolutions and microphysics schemes (**(f)**-**(l)**), and their differences (panels **(m)**-**(p)**) from the two CT case studies at the valley floor, respectively. The dashed lines denotes the average range of LIDAR observations to put the model output into context.

### 3.3 Discussion and comparison of the four cases

In general, the observations from the field campaigns proved as an excellent data pool for model validation. However, our comparison with observations deviates from classical NWP model validation methods (e.g., forecast skills, calculating bias, and root-mean square error). Therefore, most of our analysis is of qualitative matter, but still we provided an overview of the performance of the microphysics schemes and possible model shortcomings. Furthermore, we have to consider in our analysis that single point observations are not entirely representative for cloud patterns at a certain location. We tried to overcome this spatial problem by including model output from five output grid points to allow information on the spatial variability as it is represented in the model. Besides the point observations with the remote sensing systems, we provided a qualitative analysis in

comparison with all-sky camera images. The detailed comparison with the observations from measurement campaigns allowed us to perform a one-of-a-kind analysis of model cloud properties and the related physical processes.

For all four cases, the comparison with microwave radiometer and radiosonde profiles shows a realistic representation of the diurnal temperature cycle (cf. Figs. B2, B4, and C1), with only small ($< 1\,°C$) differences between horizontal resolutions, underlying terrain, and microphysics scheme. However, we noticed that the choice of the cloud microphysics scheme can impact the temperature profile in a way that we cannot fully explain. The largest differences in temperatures are notable in the $1\,km$ simulations. For example, for the case study 20 Mar 2022 we see cooler temperatures inside the cloud with 2M, while above the cloud a positive temperature anomaly is notable (Fig. B2). This is contrary to what we would normally expect: higher temperatures inside the cloud region due to latent heat release and cooler temperatures above the cloud due to cloud top cooling. This response is not present in the $65\,m$ simulations, where the temperature profiles are more realistic. We hypothesize that the reduced vertical mixing at lower resolutions (cf. Figs. 4, 8, 13) leads to an unphysical model response in the $1\,km$ simulations, however, further research is needed to investigate this model behaviour.

Regarding the cloud cover, we see that the $65\,m$ simulations achieve a better representation of the investigated cloud types albeit some shortcomings such as no full cloud cover for the case of stratocumulus clouds (31 Jan 2023) or too short-lived clouds (in all cases). There are no systematic changes when switching from 1M to 2M. For both horizontal resolutions both microphysics schemes perform similarly with respect to cloud cover occurrences. We rather see random differences between the two schemes which leads to spatial shifts of the clouds.

For the hilly terrain with a focus on wintertime clouds, it is evident that the HT1 simulations predict higher LWCs than the HT65 simulations and the observations. As expected, the IWC for HT1 was close to zero. On the other hand, the HT65 simulations qualitatively agree better with observations regarding IWC and cloud height but fail to simulate any LWC for the cases. This points towards the model struggling to represent mixed-phase clouds, with either retaining too much liquid phase or fully glaciating the clouds. A quantitative analysis showed that LWC and IWC are strongly underestimated in the model compared to the observations with 2M showing higher contents (and paths). The differences between the two horizontal resolutions are summarized in Fig. 14 for all case studies in the form of histograms. It is clear that over hilly terrain the kilometric simulations have more liquid clouds (Fig. 14a,d,j,m), while the LES simulation have more IWC and IWP (Fig. 14b,e,k,n). Moreover, the timing (20 Mar 2022) and the longevity (31 Mar 2023) still pose a challenge for the model to capture fully. For the summer cases in complex terrain, we also see that cloud formation and cloud features are better represented in the CT65 simulations compared to the CT1 runs. Nevertheless, the discrepancy to the $1\,km$ simulations is much less than for the hilly terrain; while the largest difference in LWC simulation is visible for the Aug 7 case study (altocumulus clouds, Fig. 14c,f). Interestingly, the kilometric simulations perform fairly well, especially for the case on 14 Aug 2019 (Fig. 14l,o). The better agreement between CT1 and CT65 is also noticeable in Fig. 14. Contrary to the hilly terrain simulations, the predicted LWPs for the complex terrain simulations agree reasonably well with the observations. Similar to the hilly terrain simulations, the longevity and timing of the clouds are difficult to be captured by the model, with either a too late cloud formation (5 Aug 2019) or a too early dissipation of cumulus clouds (14 Aug 2019).

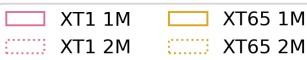

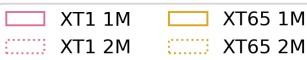

**Figure 14.** Histograms for LWC (g m$^{-3}$, bin width = 0.05 g m$^{-3}$), IWC (g m$^{-3}$, bin width = 0.05 g m$^{-3}$), LWP (g m$^{-2}$, bin width = 0.1 g m$^{-2}$), and IWP (g m$^{-2}$, bin width = 0.4 g m$^{-2}$) for all case studies sorted in hilly terrain (HT, light green) and complex terrain (CT, dark green) simulations. Similar to Table 3 and 4, a threshold value of 0.01 g m$^{-3}$ is applied before computing the histograms. Note the different scales for the x-axes.

One reason for the differences between the two horizontal resolutions is that the sub-hectometric simulations lead to a more realistic representation of vertical velocities. This leads to an earlier onset of cloud formation given that especially updrafts lead to the supersaturation required for cloud droplet and ice crystal formation. This highlights the current limitations of operational weather forecasts, that cannot resolve these small-scale vertical velocities leading to an unrealistic representation of the cloud cover. We also see that while the 1M is fairly good at capturing the different cloud types, the 2M still outperforms it in terms of cloud microphysical properties (e.g., LWC and IWC). The advantages of 2M was also discussed by Bryan and Morrison (2012) and Kovačević and Ćurić (2015), while Baldauf et al. (2011) and Kondo et al. (2021) found that 1M and 2M behave similarly. We further notice that especially for the investigated wintertime clouds over the hilly terrain the 2M performs better in representing the cloud characteristics. Over the complex terrain the differences appeared to be less significant. However, we want to highlight that 1M requires prescribed cloud droplet number concentrations, while 2M allows for variable cloud droplet number concentrations based on the updraft velocity. This is a more physical representation of the cloud formation process and should be considered when simulating clouds. Furthermore, for investigating aerosol-cloud interactions, 2M is crucial to couple the aerosol number concentrations with cloud droplet number concentrations.

## 4 Conclusions

We conducted numerical simulations with the ICON model at two horizontal grid spacings ($\Delta x = 65$ m and $\Delta x = 1$ km) for four case studies to investigate the impact of terrain, mesh size, and microphysics scheme (one-moment vs. two-moment) on the formation of two cloud types (altocumulus and stratocumulus clouds) at two locations in Europe (hilly vs. complex terrain). The simulations are validated with observations of LWC, IWC, LWP, and meteorological variables (e.g., temperature and vertical velocity) from two measurement campaigns (CLOUDLAB, hilly terrain, and CROSSINN, complex terrain). The detailed model validation study leads us to the following conclusions:

– The diurnal evolution of temperature is represented at both mesh sizes and locations in a realistic way and in qualitative agreement with observations. However, the $\Delta x = 65$ m simulations clearly outperform the kilometric simulations in terms of vertical velocity representation.

– This realistic representation of up- and downdrafts consequently leads to a better representation of clouds in the $\Delta x = 65$ m simulations in terms of cloud formation, duration, and microphysical properties.

– The cloud microphysical properties are often better represented in simulations with the two-moment scheme than compared to the one-moment scheme, which may come from the more physical representation of cloud processes such as cloud droplet formation or ice nucleation. This can be seen in the liquid and ice water content, but also in the timing of the cloud formation and the height of the clouds.

– When we compare cloud types, we note that the model performs generally better in the representation of convective clouds (cumulus and stratocumulus clouds) than for stratiform clouds (altocumulus clouds). This applies to both regions (hilly and complex terrain).

- The observations from the two independent measurement campaigns (CLOUDLAB and CROSSINN) provided a valuable dataset and were essential to validate the representation of clouds in numerical weather prediction models at high horizontal resolution.

This study provides a first evaluation of mid-level clouds over hilly and complex terrain in the form of case studies in ICON. This can be of course expanded to other cloud types and over a larger time period to increase the statistical representation. A future, more in-depth analysis is needed to include the comparison of the process rates between the two microphysics schemes. Furthermore, no sensitivity analysis was done to perturb single parameterization within the cloud microphysics schemes, which could be a further step to a more in-depth evaluation. Still, the study serves for a direct comparison of cloud representation at kilometric and sub-hectometic grid spacings with detailed observations of cloud properties, and gives a valuable overview on limitations of kilometric models for representing clouds.

*Code and data availability.* Observational data from the CROSSINN campaign (Adler et al., 2021b) can be downloaded from Adler et al. (2021a) [radiosondes, all-sky camera, ceilometer, and HATPRO] and Gohm et al. (2021) [LIDAR data]. Model and observational data and analysis and plotting scripts are available upon request and will be made publicly available upon acceptance of the manuscript. We used the ICON model code version 2.6.6 for our simulations. The open-source model code can be obtained at https://icon-model.org/.

*Video supplement.* The video compilations for the all-sky cameras and model cloud cover can be found at Omanovic et al. (2024b).

**Appendix A: Distribution of model levels**

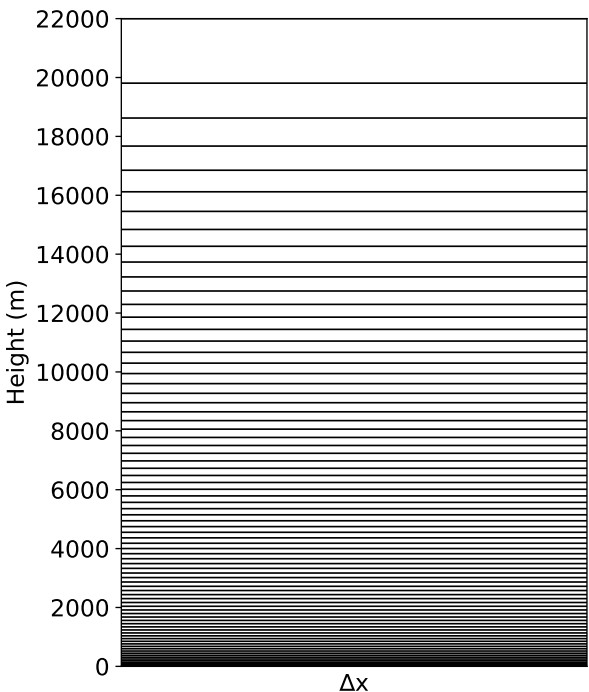

**Figure A1.** Mean height of the 80 model levels with model top at 22000 m valid for all simulations.

## Appendix B: Additional figures for HT simulations

### B1    20 Mar 2022: Altocumulus clouds

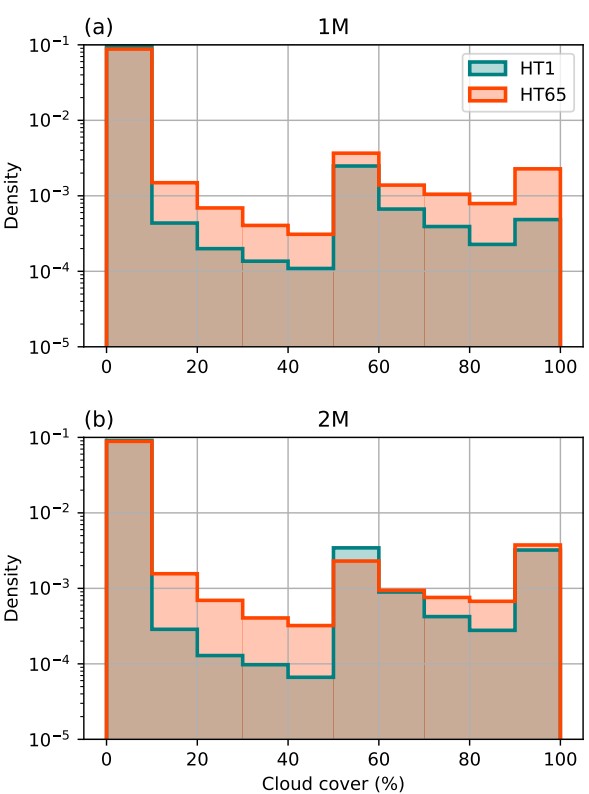

**Figure B1.** Probability density figures for the cloud cover (%) of HT1 (teal) and HT65 (red) with 1M (**a**) and 2M (**b**) over all five minute model output time steps and all height levels, where the cloud was present. The domain and time period included in this analysis is the same as in Fig. 3. The bin width is set to 10 %.

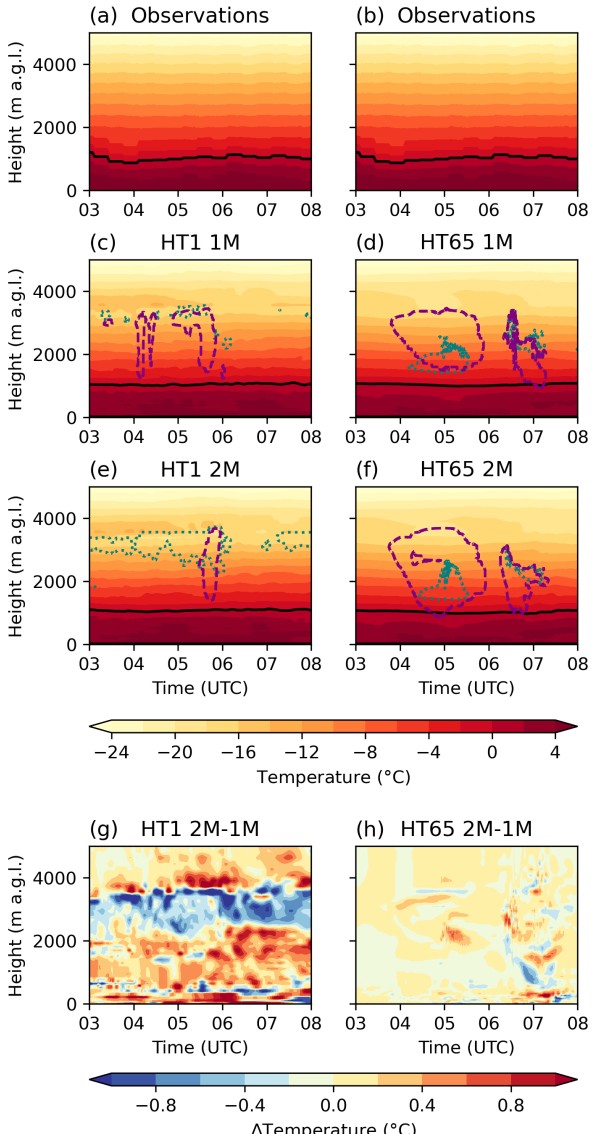

**Figure B2.** Case study 20 Mar 022: Hovmöller diagrams for temperature (°C) measured by a microwave radiometer (**a, b**) and simulated by the model for both resolutions (HT1 and HT65) and both cloud microphysics schemes (1M and 2M) (**c-f**).The black line indicates the 0 °C isotherm (black line). The dashed purple line shows the IWC $= 0.01$ g m$^{-3}$, while the dotted teal line shows the LWC $= 0.01$ g m$^{-3}$. These serve as an indicator for the position of the cloud. The last row (**g**, **h**) shows the differences between 2M and 1M for each resolution.

## B2    31 Jan 2023: Stratocumulus clouds

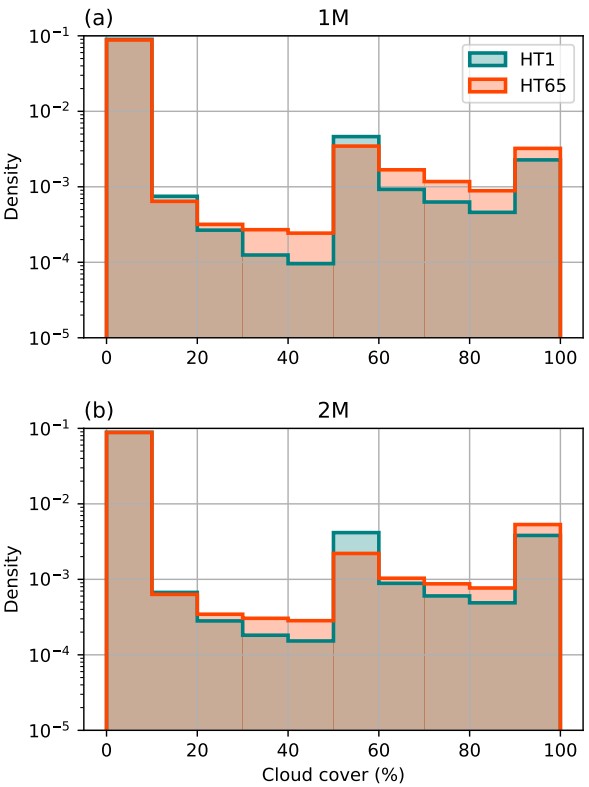

**Figure B3.** Probability density figures for the cloud cover (%) of HT1 (teal) and HT65 (red) with 1M (**a**) and 2M (**b**) over all five minute model output time steps and all height levels, where the cloud was present. The domain and time period included in this analysis is the same as in Fig. 7. The bin width is set to 10 %.

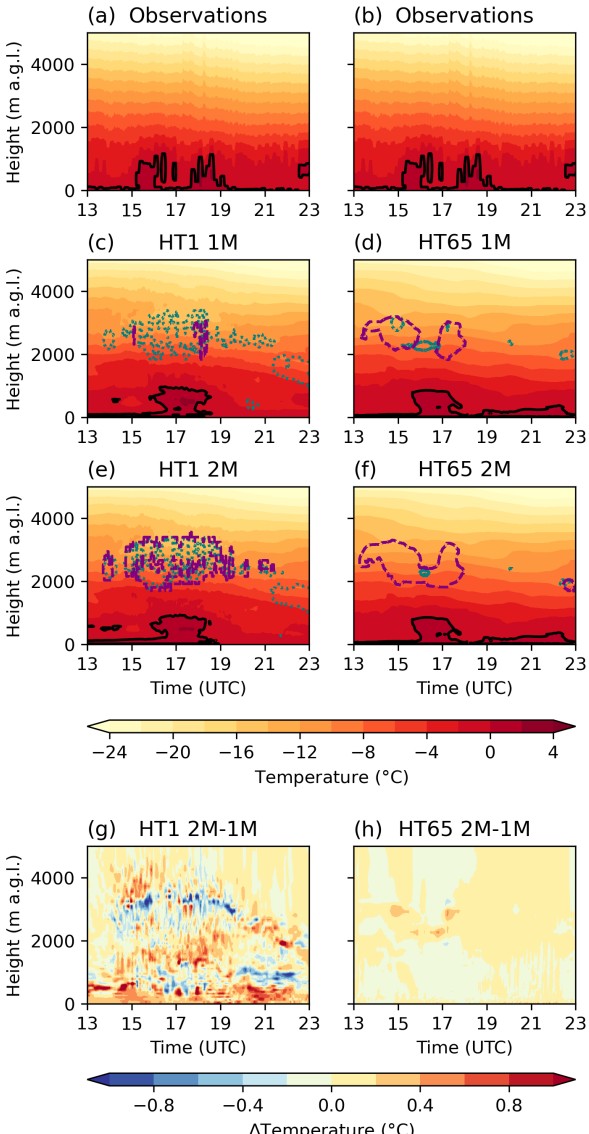

**Figure B4.** Case study 31 Jan 2023: Hovmöller diagrams for temperature (°C) measured by a microwave radiometer (**a, b**) and simulated by the model for both resolutions (HT1 and HT65) and both cloud microphysics schemes (1M and 2M) (**c-f**). The black line indicates the 0 °C isotherm (black line). The dashed purple line shows the IWC $= 0.01$ g m$^{-3}$, while the dotted teal line shows the LWC $= 0.01$ g m$^{-3}$. These serve as an indicator for the position of the cloud. The last row (**g, h**) shows the differences between 2M and 1M for each resolution.

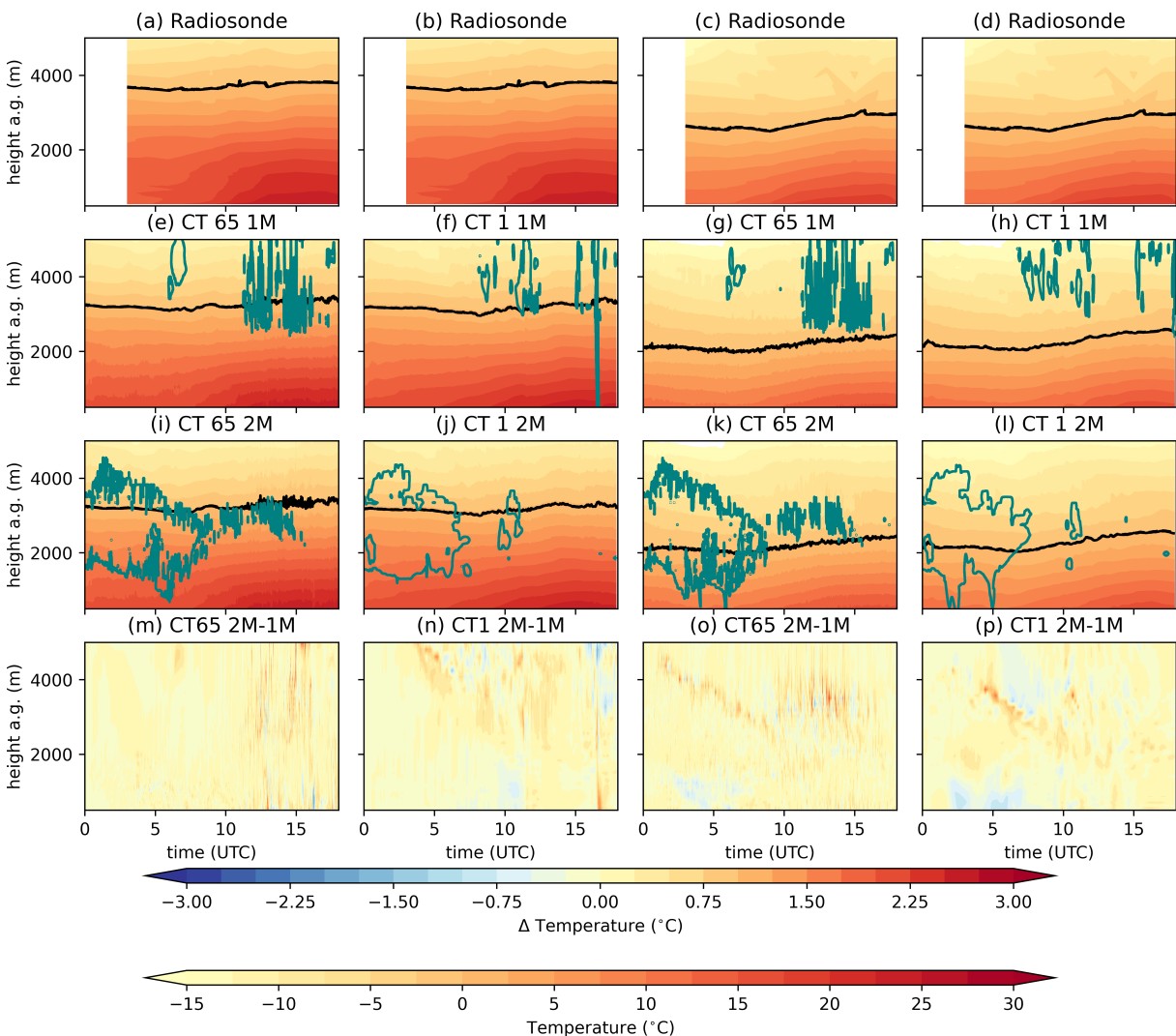

**Figure C1.** Time evolution of vertical profiles of air temperature from radiosonde observations (panels **(a)**-**(d)**), and model output from the different horizontal resolutions and microphysics schemes (**(f)**-**(l)**), and their differences (panels **(m)**-**(p)**) from the two CT case studies (05 Aug 2019, two left columns; and Aug 14, 2029, two right columns) at the valley floor, respectively. The black lines denotes the $0\,°C$ line, and the green contours show areas where LWC $>0.01\,g\,m^3$.

# C1    05 Aug 2019: Altocumulus clouds

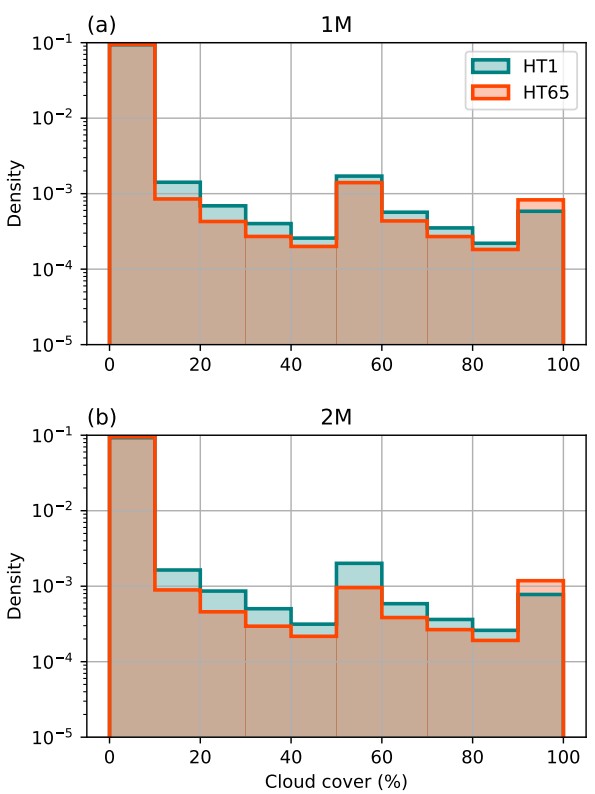

**Figure C2.** Probability density figures for the cloud cover (%) of HT1 (teal) and HT65 (red) with 1M (**a**) and 2M (**b**) over all five minute model output time steps and all height levels, where the cloud was present. The domain and time period included in this analysis is the same as in Fig. 9. The bin width is set to 10 %.

## C2   14 Aug 2019: Altocumulus clouds

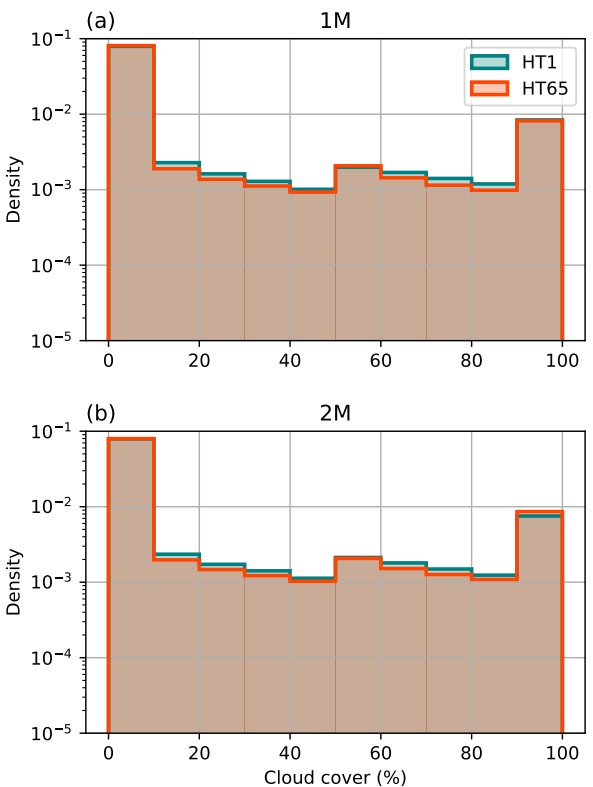

**Figure C3.** Probability density figures for the cloud cover (%) of HT1 (teal) and HT65 (red) with 1M (**a**) and 2M (**b**) over all five minute model output time steps and all height levels, where the cloud was present. The domain and time period included in this analysis is the same as in Fig. 11. The bin width is set to 10 %.

## Appendix D: Hydrometeor number concentrations

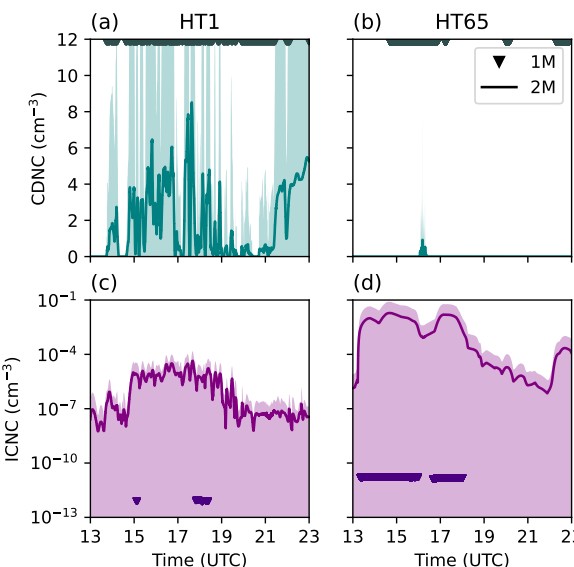

**Figure D1.** Simulated cloud droplet (**a** and **b**) and ice crystal (**c** and **d**) number concentrations (CDNC and ICNC, respectively) for 31 Jan 2023 for both resolutions (HT1 and HT65, **a/c** and **b/d**, respectively). The number concentration from 1M are shown as markers, whenever there is a cloud present. In 1M CDNC = $200\,\mathrm{cm}^{-3}$ (prescribed, markers only for indicating cloudy conditions), while ICNC is calculated as a function of temperature following Cooper (1986) (diagnosed). The predicted quantities from 2M are shown as means (solid lines) $\pm$ standard deviations (shadings).

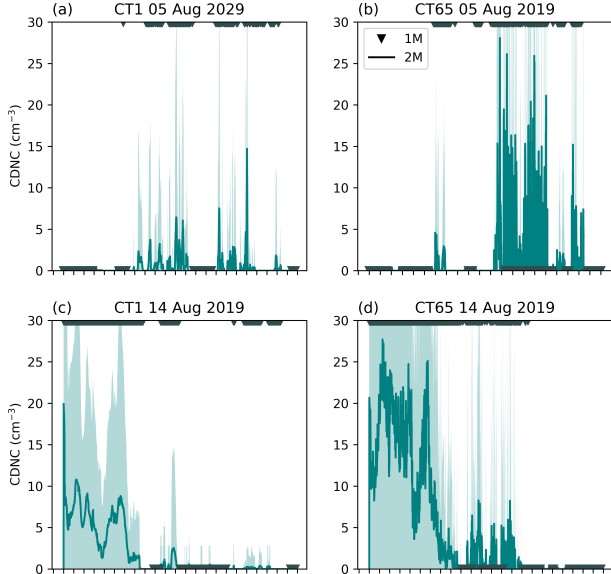

**Figure D2.** Simulated cloud droplet number concentrations (CDNC) for 5 Aug (**a**, **b**) and 14 Aug (**c**, **d**) for both resolutions (CT1 and CT65, **a/c** and **b/d**, respectively). The number concentration from 1M are shown as markers, whenever there is a cloud present. In 1M CDNC = 200 cm$^{-3}$ (prescribed, markers only for indicating cloudy conditions). The predicted quantities from 2M are shown as means (solid lines) $\pm$ standard deviations (shadings).

*Author contributions.* NO conceived the study idea, and NO and BG designed and conducted the hilly terrain (NO) and complex terrain (BG) simulations, respectively. Both authors wrote the manuscript and performed model output analysis and observations. UL provided input to the manuscript writing and results discussion.

*Competing interests.* The authors declare no competing interest.

*Acknowledgements.* The CLOUDLAB project (NO) has received funding from the European Research Council (ERC) 411 under the European Union's Horizon 2020 research and innovation program (grant agreement 412 No. 101021272 CLOUDLAB). This work was supported by a grant from the Swiss National Supercomputing Centre (CSCS) under project ID s1144 (NO) and d121 (BG). BG is funded by EX-CLAIM, a project by ETH Zurich. We thank the two anonymous referees for providing thoughtful comments leading to an improvement of the manuscript.

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
