# Peer review of "The impact of mesh size and microphysics scheme on the representation of mid-level clouds in the ICON model in hilly and complex terrain"

_EGUsphere, 2024_

## Referee Comment (RC1)

**Review of "The impact of mesh size and microphysics scheme on the representation of mid-level clouds in the ICON model in hilly and complex terrain" by Omanovic et al. (egusphere-2024-1989)**

This manuscript addresses how the ICON model can represent clouds at different horizontal resolutions (1000 and 65 m) and with different cloud microphysical schemes (one- and two-moment microphysics), re-simulating in total for cases originating from observational campaigns in Switzerland and Austria. Considering the increasing use of high-resolution models in weather and climate prediction, this study touches upon an important subject, but does not substantiates its claims. To strengthen the manuscript, I believe that a much deeper analysis of the effects of resolution and cloud microphysical schemes are necessary. Thus, I cannot recommend the publication in the current stage.

**Major Comments**

*Constraining the influence of the large-scale model.* I understand that all simulations are driven by COSMO-1 analysis (ll. 114 – 116). How much does this large-scale forcing affect the simulations, and especially the development of clouds? In other words, how much alike are thermodynamic properties in the COSMO-1 driver model and the corresponding ICON runs? Besides the analyzed impact of vertical velocity variances, the ability of the nested models to change the underlying thermodynamics is probably key to understand how resolution and cloud microphysics can affect the simulated scenarios.

*Extend the analysis of cloud microphysical properties.* The authors restrict their analysis of cloud microphysical properties to liquid and ice masses. While I understand that liquid and ice masses are predicted by both microphysical schemes, addressing how well droplet and ice concentrations are predicted in the two-moment and prescribed in the first-moment scheme is crucial to evaluate how well cloud microphysics are captured. Moreover, the direct comparison of process rates (e.g., condensation/evaporation, accretion) could enable more detailed insights on how the cloud microphysical schemes behave.

*Numerical effects.* The authors did not address the effects of numerics. Increasing the resolution from 1000 to 65 m should go along with a substantial decrease in numerical diffusion. Simultaneously, approximately doubling the number of predicted moments in the cloud microphysics scheme will do the opposite, i.e., potentially increasing the numerical diffusion of cloud properties.

**Minor Comments**

L. 102: What is meant by "peak-to-peak-distance"? The width of the valley?

Ll. 111 – 112: Both ICON resolutions use the same 80 vertical levels. How much are these levels apart? Why did the authors not address the impact of vertical resolution, which probably has a strong impact on vertical velocities and hence cloud microphysics.

Ll. 126 – 163: The description of cloud microphysics combines the underlying physics and the applied models in a slightly unfortunate way, making it hard for the reader to disentangle what processes are actually represented in the cloud microphysical model. For instance, most activation parameterizations do not depend on a "critical size" (l. 134), but only the critical supersaturation. Moreover, most models do not represent hygroscopic growth of aerosols (l. 132). Is it resolved in the applied models? Lastly, do the models use saturation adjustments for their treatment of condensation? If yes, this needs to be mentioned, and potential impacts on the simulated cloud microphysics should be discussed.

Ll. 174 – 178: Please elaborate on the categorization of altocumulus, stratocumulus, and cumulus. Why are these clouds considered "mid-level" clouds, although cumulus and stratocumulus are typically considered low-level boundary-layer clouds?

Ll. 187 ff.: How is "cloud cover defined"? Figure 2 states that the cloud cover is derived from a certain height level. Many authors use an integrated quantity (e.g., cloud optical thickness) to define cloud cover. What are the benefits in defining the cloud cover in the applied way?

Appendices: There are many nice figures in the appendices. What about integrating them as sub-panels into the main text?

Tabs. 3 ff.: Why do the authors use LWC and IWC and not integrated quantities such as LWP and IWP? The former might be more affected by subtle differences in cloud depth.

Ll. 192 – 194: Largest differences at the cloud edge might hint toward differences in turbulent (or numerical) mixing/diffusion. This should be addressed more thoroughly.

L. 196: What is "simulation 3"?

Ll. 214 – 216: A stratocumulus of 2000 m depth is rather unusual.

Figs. 7 and 9: What do the dashed lines in panels b to f indicate?

Figs. A2, A5, B1: These plots would benefit from a clear indication of the location of the cloud.

**Technical Comments**

Ll. 80 – 82: The campaigns are probably not "completely independent of each other". I suggest rewording.

L. 84: Change "case studies" to "cases simulated here".

L. 269: "LWP" has already been introduced as an abbreviation for "liquid water path". Use it.

Ll. 273 ff.: Semicola are overused in the text.

L. 286: "Ac" is not defined.

L. 314: To what does the "at" point to?

---

## Referee Comment (RC2)

Comments on "The impact of mesh size and microphysics scheme on the representation of mid-level clouds in the ICON model in hilly and complex terrain" by Omanovic et al.,

This paper presents a study using numerical simulations with the ICON model at two horizontal grid spacings (65m and 1km) to investigate the effects of terrain, mesh size, and microphysics schemes (one-moment vs. two-moment) on cloud formation in two European regions (hilly and complex terrain). Four case studies were conducted, validated against observations of liquid water content (LWC), ice water content (IWC), liquid water path (LWP), and meteorological variables collected during the CLOUDLAB (hilly terrain) and CROSSINN (complex terrain) campaigns. This study is a valuable first evaluation of mid-level cloud simulations over complex terrain using ICON and highlights limitations of kilometric models in representing clouds, suggesting future work on sensitivity analysis and expanding cloud-type representation.

Overall, I think this is very solid and well-written manuscript. I suggest publish it with minor corrections.

Minor comments:
- Caption of Figure 3, 5: ()-(), double right bracket
- The solid and dash lines are not clear to differential each other in Figure 9j.

---

## Author Response (AR1)

**Response to Referees**
**The impact of mesh size and microphysics scheme on the representation of mid-level clouds in the ICON model in hilly and complex terrain**

Nadja Omanovic, Brigitta Goger, and Ulrike Lohmann

September 26, 2024

Dear editor and referees,

We would like to thank the editor for handling our manuscript and the referees for their careful evaluation of the revised manuscript. Below we address our detailed responses to all the comments. In this response-to-review document we try to clarify and address each of the suggestions, comments and questions made during the review process. Therefore we have copied the comments in blue boxes and have addressed them one by one. In the response we use italic fonts to quote text from the revised manuscript. Additional to the revised manuscript, we have uploaded a version of the manuscript with highlighted tracked changes).

Best regards, Nadja Omanovic, Brigitta Goger, and Ulrike Lohmann

**Response to referee #1**

**Comments to the Author**

> This manuscript addresses how the ICON model can represent clouds at different horizontal resolutions (1000 and 65 m) and with different cloud microphysical schemes (one- and two-moment microphysics), re-simulating in total four cases originating from observational campaigns in Switzerland and Austria. Considering the increasing use of high-resolution models in weather and climate prediction, this study touches upon an important subject, but does not substantiates its claims. To strengthen the manuscript, I believe that a much deeper analysis of the effects of resolution and cloud microphysical schemes are necessary. Thus, I cannot recommend the publication in the current stage.

We thank the referee for the constructive comments and we will address the concerns below.

**Major comments**

> **Constraining the influence of the large-scale model.** I understand that all simulations are driven by COSMO-1 analysis (ll. 114 – 116). How much does this large-scale forcing affect the simulations, and especially the development of clouds? In other words, how much alike are thermodynamic properties in the COSMO-1 driver model and the corresponding ICON runs? Besides the analyzed impact of vertical velocity variances, the ability of the nested models to change the underlying thermodynamics is probably key to understand how resolution and cloud microphysics can affect the simulated scenarios.

Davies (2014) found that close similarity between the driving and the nested model is beneficial for representing meteorological fields in the nested model. They also note that the driving model only makes up a fraction of the total model error. The same strategy was used by Heinze et al. (2017), who drove two LES models with realistic input data from NWP models to simulate cloud properties over Germany. Schemann et al. (2020) showed with a similar LES setup (with the ICON model) that the driving model had an influence on cloud formation in their innermost LES domains. Their best choice for a driving model to simulate cloud properties in the LES domain was indeed boundary data from the COSMO-DE model, a NWP model. This strategy was continued by Schemann and Ebell (2020) to simulate mixed-phase clouds in the Arctic. Therefore, we decided to use the same strategy with the choice of our driving model (i.e., the COSMO-1 analysis). We added following in the model setup description (line

123):
"*This setup is similar to Schemann et al. (2020), who found that constraining the model with a driving model (COSMO-1) yields an improved model performance compared observations of cloud properties. Furthermore, the impact of the driving model on the nested model should be minimal compared to the internal model variability and errors (Davies, 2014).*"

The COSMO-1 model was the operational NWP model of the Swiss Federal Office for Meteorology and Climatology (MeteoSwiss) during our time period of interest (i.e., 2019 and 2023) and the predecessor of ICON. While the dynamical core was rewritten, the physics remained fairly similar expect for the radiation scheme (switch from RRTM (Mlawer et al.; Barker et al., 2003) to ecRad (Hogan and Bozzo, 2018)). Another option for a driving model would have been the IFS from ECMWF, which has the same radiation scheme but very different physics and a much coarser resolution (9 km). The COSMO-1 analyses are the only high-resolution analysis data over the Alps available to us, and we decided to use COSMO-1 analyses as a driver to have the closest possible ground truth from a modelling perspective.

We include here a figure of the COSMO analysis for each case study during the cloud event showing the integrated cloud cover over mid levels (400 - 800 hPa) and the corresponding 1 km simulation in ICON (Fig. R1 and R2). We see that the COSMO-1 analysis reproduces the discussed clouds in a realistic way with either full cloud coverage (altocumulus clouds) or more patchy clouds showing the transition to more convective clouds observed in Austria. These patterns can also be seen in the ICON 1 km simulations (and the two microphysics schemes differ), but we note that we have a comparable situation between COSMO-1 and ICON-1. We added the following sentence to the manuscript (Section 2.2, line 121):

*A brief comparison of cloud representation in the COSMO-1 analysis and the resulting ICON runs ($\Delta x{=}1\,km$) revealed realistic cloud patterns in the COSMO analysis data and no large discrepancies with ICON.*

[Figure]

Figure R1: COSMO-1 analyses for integrated (400 - 800 hPa) cloud cover for all four case studies at selected time steps.

[Figure]

(a) Case studies in hilly terrain.

(b) Case studies in complex terrain.

Figure R2: ICON 1 km simulations with integrated (400 - 800 hPa) cloud cover for all four case studies and the two microphysics schemes (1M and 2M).

> **Extend the analysis of cloud microphysical properties.** The authors restrict their analysis of cloud microphysical properties to liquid and ice masses. While I understand that liquid and ice masses are predicted by both microphysical schemes, addressing how well droplet and ice concentrations are predicted in the two-moment and prescribed in the first-moment scheme is crucial to evaluate how well cloud microphysics are captured. Moreover, the direct comparison of process rates (e.g., condensation/evaporation, accretion) could enable more detailed insights on how the cloud microphysical schemes behave.

Our main focus was to validate ICON's performance with available observational data of cloud properties (i.e., LWC, IWC, LWP, ceilometer observations, etc), because we initially thought that this analysis would also give insight into the performance of the two microphysics schemes.

However, we agree with the referee that a more in-depth analysis of the microphysics scheme would strengthen our conclusions. Unfortunately, we do not have any measurements regarding the number concentrations of the hydrometeors, which makes it difficult to discuss how each scheme performs. We are not including process rates because we are unfortunately constrained by a current transition of our computing centre (CSCS) to a new machine, so we cannot re-run the present simulations to diagnose process rates. However, we mention in the outlook of the manuscript that a detailed analysis of the process rates will be necessary for a future, more in-depth validation of the microphysics schemes (line 450):*A future, more in-depth analysis would also need to include the comparison of the process rates between the two microphysics schemes.*.

We added a discussion and the accompanying figures of the CDNC and ICNC values in Section 3.1.1. (line 233): "*Figure 5 (for the second case study: see Appendix D, Fig. D1, for the simulations in complex terrain see Fig. D2) shows the temporal evolution of the cloud droplet and ice crystal number concentrations (CDNC and ICNC, respectively) for both resolutions and both microphysics schemes for the hilly terrain case studies. As mentioned, the CDNC are prescribed in 1M ($=200\,cm^{-3}$) as soon as a cloud forms. In the model CDNC is only used for calculating the collection kernels between the droplets, and it does not change over time. In the figures, the markers are only set for illustrative purposes highlighting when the LWC exceeded $0.01\,gm^-3$ in 1M simulations. In 2M, CDNC are predicted, and thus change with time and depend on the cloud droplet activation and removal due to collision processes (see Sect. 2). We see a large discrepancy between the two microphysics schemes, with the prescribed concentration in 1M probably being a better estimate for the CDNC for a continental cloud, while 2M predicts rather low concentrations. This could be a consequence of interactions with the ice phase, where at subzero temperatures, the ice phase is the favored state, and thus ice crystals will form and grow at the expense of evaporating cloud droplets. One hypothesis is, that this balance in the model is more on the side of the ice crystals. This is further supported by the strong underestimation of the LWC/LWP in the simulations. For ICNC we see that 1M strongly underestimates it,which may arise from the equation for ICNC from Cooper (1986), where at temperatures around -10 °C the ICNC activity is underestimated. For 2M we see that only for the HT65 simulation a realistic ICNC is simulated with concentrations maximizing at $0.1\,cm^{-3}$, while for HT1 ICNC is by almost three magnitudes of order too low, also almost no IWC/IWP was simulated. Hence, while the CDNC is strongly underestimated in 2M simulations, which may come from the balance between the liquid and ice phase. This is an issue weather and climate models struggle with (Liu et al., 2011; Kay et al., 2016; Klaus et al., 2016; McIlhattan et al., 2017; Kretzschmar et al., 2019; Huang et al., 2021; Omanovic et al., 2024), we see a more realistic simulation of ICNC than for 1M. For the complex terrain case studies in summer, we only investigate CDNC (Fig. D2). The concentrations are slightly higher (factor 2) than for the HT simulations, but this is probably still an underestimation as higher CDNC over land can be expected (Lohmann et al., 2016).*"

> **Numerical effects.** The authors did not address the effects of numerics. Increasing the resolution from 1000 to 65 m should go along with a substantial decrease in numerical diffusion. Simultaneously, approximately doubling the number of predicted moments in the cloud microphysics scheme will do the opposite, i.e., potentially increasing the numerical diffusion of cloud properties.

The increase in horizontal resolution does not necessarily lead to a complete decrease in diffusion, because we note that with higher resolution, roughness elements, and especially in our case, topography, is better resolved and henceforth leads to more mechanisms of turbulence generation. To show this, we computed the power spectral density of selected variables in our simulations in Fig. R3. The figures suggest that the HT65 and CT65 simulations are able to resolve more scales than their 1 km counterparts, and furthermore, the power spectra in the LES range follow Kolmogorov's -5/3 slope (especially visible for vertical velocity). Changing the microphysics scheme has a negligible impact on the energy spectra. Henceforth, we conclude that turbulence/diffusion is represented in a more realistic way in the 65 m simulations.

Furthermore, we discussed the diffusivity of the microphysics schemes with Axel Seifert (author of the one and

two moment microphysics schemes in ICON Seifert and Beheng (2006)), and having additional tracers should not impact the diffusivity of the schemes as the diffusivity originates from the advection scheme (Axel Seifert, personal communication).

[Figure]

(a) Energy spectra for hilly terrain simulations for variables U (horizontal wind component), W (vertical velocity), and T (temperature).

[Figure]

(b) Energy spectra for complex terrain simulations for variables U (horizontal wind component), W (vertical velocity), and T (temperature).

Figure R3: Energy spectra for 1 km and 65 m simulations with 1M and 2M, respectively.

**Minor comments**

L. 102: What is meant by "peak-to-peak-distance"? The width of the valley?

Exactly - the valley width between the two highest peaks along a cross-section, while the valley width at the valley floor is around 5 km. We changed the sentence to (line 103):
*The Inn Valley is a major east-to-west oriented Alpine valley with a peak-to-peak distance of 10 km, while the valley bottom extends 5 km at our location of interest [...]*

Ll. 111 – 112: Both ICON resolutions use the same 80 vertical levels. How much are these levels apart? Why did the authors not address the impact of vertical resolution, which probably has a strong impact on vertical velocities and hence cloud microphysics.

We focus here on a widely used setup (e.g., MeteoSwiss and in Goger and Dipankar (2024); Omanovic et al. (2024)). Furthermore, the impact of vertical resolution was tested for different cases from the ones shown here and we did not notice any improvement in cloud representation or temperature inversions. We added a figure in the appendix illustrating the mean distribution of the vertical levels in the model (line 113):
*"[...] and 80 vertical levels with the model top at 22 km (see Fig. A1). A recent study by Schmidt et al. (2024)*

*showed that for varying vertical resolutions (on a global storm-resolving scale, i.e. 5 km) no convergence could be found for the microphysical properties. Also, tests conducted with the model setup (but for different case studies) showed no improvement of cloud representation or temperature inversions with increasing vertical resolution. Hence, we decided to only vary the horizontal resolution, and keep the vertical resolution the same as it is used by the Swiss Federal Office of Meteorology (MeteoSwiss)".*

Ll. 126 – 163: The description of cloud microphysics combines the underlying physics and the applied models in a slightly unfortunate way, making it hard for the reader to disentangle what processes are actually represented in the cloud microphysical model. For instance, most activation parameterizations do not depend on a "critical size" (l. 134), but only the critical supersaturation. Moreover, most models do not represent hygroscopic growth of aerosols (l. 132). Is it resolved in the applied models? Lastly, do the models use saturation adjustments for their treatment of condensation? If yes, this needs to be mentioned, and potential impacts on the simulated cloud microphysics should be discussed.

Thank you for your comment. We tried to rephrase and restructure that part by introducing a list for the single processes. We also included the description of the saturation adjustment. We changed it to the following (line 139):"

[revised manuscript text omitted]

> Ll. 174 – 178: Please elaborate on the categorization of altocumulus, stratocumulus, and cumulus. Why are these clouds considered "mid-level" clouds, although cumulus and stratocumulus are typically considered low-level boundary-layer clouds?

We identified the cloud types based on their height but also the boundary layer height which was inferred from the aerosol layer height measured by a ceilometer. For the case on 20 Mar 2022 and 5 Aug 2019, which we classified as altocumulus, we see in Fig. 3a/c and 7a that the cloud is clearly "decoupled" from the aerosol layer, thus a clear mid-level cloud. This was not the case for 31 Jan 2023. However, based on your comment, we adapted the classification for 14 Aug 2019, because based on our classification the clouds first classified as cumulus clouds are actually decoupled from the aerosol layer height, and thus we classify them now as "altocumulus clouds". We adapted following in Section 2.3 "Case study selection" (line 188):
" *Hence, we decided to focus on the following cloud types: altocumulus (Ac, see Fig. 1f, h, and i) and stratocumulus (Sc, see Fig. 1g) clouds. We identified the cloud types based on their height and the aerosol layer height, which serves as an indicator for the boundary layer height (see Fig. 3a and c, 5a and c). If the cloud was above the aerosol layer height, we classified it as an Ac cloud, otherwise as a Sc cloud.*".

> Ll. 187 ff.: How is "cloud cover defined"? Figure 2 states that the cloud cover is derived from a certain height level. Many authors use an integrated quantity (e.g., cloud optical thickness) to define cloud cover. What are the benefits in defining the cloud cover in the applied way?

The cloud cover is a diagnostic in the model based on the LWC and IWC in that grid box. We chose the cloud cover instead of a cloud optical depth to have a better comparability to the Allskycamera snapshots. The height was chosen such that the cross section went through the "middle" of the cloud. From the appended histograms (in the appendices) for the cloud cover, we see that the simulations barely differ in any regard. We added following sentence to the paragraph (line 207): "*The cloud cover is a diagnostic based on the present LWC and IWC in the model grid box and ranges from 0 to 100 %.*"

> Appendices: There are many nice figures in the appendices. What about integrating them as sub-panels into the main text?

Thank you for the suggestion, we added the figures showing the vertical velocities in the main text as they have a pivotal impact on cloud representation in our simulations.

> Tabs. 3 ff.: Why do the authors use LWC and IWC and not integrated quantities such as LWP and IWP? The former might be more affected by subtle differences in cloud depth.

We now also added the LWP and IWP in the tables to have a full overview of all discussed quantities.

> Ll. 192 – 194: Largest differences at the cloud edge might hint toward differences in turbulent (or numerical) mixing/diffusion. This should be addressed more thoroughly.

Please see our answer on this issue in the response to the major comments.

> L. 196: What is "simulation 3"?

Thank you for finding this. The number 3 referred to the figure. We have adapted this (line 215):
"*[…]in the observations and the model simulations (Fig. 3).*".

**Ll. 214 – 216: A stratocumulus of 2000 m depth is rather unusual.**

We rephrased the sentence including the hypothesis that the possible sedimentation of ice crystals (long purple streaks in Fig. 5c) are misleading in classifying the cloud depth. Hence, we softened the statement (line 256):
"*The cloud lived for several hours and at times ice crystals sedimented towards the ground giving a large vertical extent to the cloud (see Fig. 7a and c).*".

**Figs. 7 and 9: What do the dashed lines in panels b to f indicate?**

The dashed lines in Figs. 7 and 9 represent the time of the vertical profiles shown in panels (h) and (j). We added the explanation to the caption of Figs. 7 and 9.

**Figs. A2, A5, B1: These plots would benefit from a clear indication of the location of the cloud.**

Thank you for the idea! We added the LWC and IWC (where applicable) with a threshold of 0.01 g m$^{-3}$ to the subfigures showing the absolute values. We did not overlay them for the difference figures to not obscure the temperature values.

**Technical comments**

Ll. 80 – 82: The campaigns are probably not "completely independent of each other". I suggest rewording.

Thank you, we reworded as follows (line 80): "*The two field campaigns were conducted independently of each other and with different research foci (cloud and boundary layer research, respectively) at two different locations (hilly terrain in the Swiss Alpine foreland and highly complex terrain in the Austrian Alps, respectively).*"

L. 84: Change "case studies" to "cases simulated here".

We changed this, thank you.

L. 269: "LWP" has already been introduced as an abbreviation for "liquid water path". Use it.

We adapted it where necessary, thank you.

Ll. 273 ff.: Semicola are overused in the text.

Thank you, we adapted the affected sentences.

L. 286: "Ac" is not defined.

We now defined the acronyms of the cloud types in Section 2.3 "Case study selection" (line 177), thank you.

L. 314: To what does the "at" point to?

The brackets around "$\approx 06{:}00\,\text{UTC}$" were not needed, and we removed them, thank you.

**Response to referee #2**

Comments on "The impact of mesh size and microphysics scheme on the representation of mid-level clouds in the ICON model in hilly and complex terrain" by Omanovic et al.,

This paper presents a study using numerical simulations with the ICON model at two horizontal grid spacings (65m and 1km) to investigate the effects of terrain, mesh size, and microphysics schemes (one-moment vs. two-moment) on cloud formation in two European regions (hilly and complex terrain). Four case studies were conducted, validated against observations of liquid water content (LWC), ice water content (IWC), liquid water path (LWP), and meteorological variables collected during the CLOUDLAB (hilly terrain) and CROSSINN (complex terrain) campaigns. This study is a valuable first evaluation of mid-level cloud simulations over complex terrain using ICON and highlights limitations of kilometric models in representing clouds, suggesting future work on sensitivity analysis and expanding cloud-type representation.

Overall, I think this is very solid and well-written manuscript. I suggest publish it with minor corrections.

Thank you for the very positive comments on our manuscript. We will change the minor comments (see below) accordingly in the revised manuscript.

Caption of Figure 3, 5: ()-(), double right bracket

Changed accordingly.

The solid and dash lines are not clear to differential each other in Figure 9j.

Thank you for the remark. We now changed the line colors to make the lines more distinguishable in Figs. 10 and 12.

---

## Referee Report (RR1)

**Review of "The impact of mesh size and microphysics scheme on the representation of mid-level clouds in the ICON model in hilly and complex terrain" by Omanovic et al. (egusphere-2024-1989)**

The authors addressed my comments with great care, and their additional explanations provided important insights. Thank you! I only have two minor comments left below.

**Minor Comments**

Tabs. 3 and 4: Could the authors comment on the assumed 'cloud mask' they used to determine the LWC, IWC, LWP, and IWP? Since most modeled mean values are substantially smaller than the observed values, it might be the case that more almost non-cloudy columns are considered in the calculation of the modeled mean values. Since the LWC also strongly depends on the height above the cloud base, I wonder at which height level the LWC is determined. Lastly, I suggest showing histograms of LWC, IWC, LWP, and IWP.

Fig. 5: Since the two-moment microphysics predict very low droplet concentrations, I wonder if autoconversion/accretion are affecting these simulations. Those processes should be absent in the one-moment microphysics simulations.

---

## Author Response (AR2)

**Response to referee and editor**
**The impact of mesh size and microphysics scheme on the representation of mid-level clouds in the ICON model in hilly and complex terrain**

Nadja Omanovic, Brigitta Goger, and Ulrike Lohmann

October 31, 2024

Dear editor and referee,

We would like to thank the editor and the referee for their careful evaluation of the revised manuscript. We address the raised points below.

Best regards, Nadja Omanovic, Brigitta Goger, and Ulrike Lohmann

**Response to referee**

**Comments to the Author**

> The authors addressed my comments with great care, and their additional explanations provided important insights. Thank you! I only have two minor comments left below.

Thank you for carefully re-reading our revised manuscript.

**Minor comments**

> Tabs. 3 and 4: Could the authors comment on the assumed 'cloud mask' they used to determine the LWC, IWC, LWP, and IWP? Since most modeled mean values are substantially smaller than the observed values, it might be the case that more almost non-cloudy columns are considered in the calculation of the modeled mean values. Since the LWC also strongly depends on the height above the cloud base, I wonder at which height level the LWC is determined. Lastly, I suggest showing histograms of LWC, IWC, LWP, and IWP.

Thank you for your question. Double-checking our analysis lead to the finding of a missing mask in the analysis of the simulated water contents and paths. We applied now a mask of LWC / IWC $> 0.01\,\mathrm{g\,m^{-3}}$ (as we have done already for the observations). We included now that the values are based on a mask in the table captions as well as in the methods part (line 189): *"For comparing the cloud characteristics, we apply a threshold value of $0.01\,\mathrm{g\,m^{-3}}$ for liquid and ice water content to the observations and simulations to only consider in-cloud values."*. With the mask, we obtain a better agreement between the observed and simulated values. We adapted the discussion as well to account for that. LWC and IWC are calculated for the entire cloud not at a specific height, we also clarified that in the table captions.

Thank you for suggesting to add histograms of the water quantities. We created a summary figure (now Fig. 14) depicting the histograms and included it in the discussion as it provides a great summary of the differing performances of the model setups.

> Fig. 5: Since the two-moment microphysics predict very low droplet concentrations, I wonder if autoconversion/accretion are affecting these simulations. Those processes should be absent in the one-moment microphysics simulations.

Autoconversion/ accretion are present both in 1M and 2M. The 1M takes the prescribed cloud droplet number concentration (or diagnosed ice crystal number concentration), and calculates the collection kernels. In 2M, this is done more realistically, because of the prognostic cloud droplet and ice crystal number concentration. In 1M, the autoconversion rate should be small given the low simulated LWC and higher cloud droplet number concentration,

hence the cloud droplet sizes are so small that autoconversion is not efficient. For 2M we have low cloud droplet number concentrations and low LWC, hence also here the autoconversion rate is small. We added the following text to the discussion for Fig. 5 (line 249): *"In both schemes, the collision and coalescence of cloud droplets (i.e., autoconversion) may be low. In 1M the high CDNC and low LWC lead to small cloud droplets limiting collisions and in 2M low CDNC and low LWC also yield small autoconversion rates."*.

**Response to editor**

**Comments to the Author**

My further request is to add units of LWP/IWP to the Figures 3 and 7 (or captions). Please also double check the observed LWC of 2.40 and LWP of 0.17 in Table 3, one of which looks like a mistake.

Thank you for spotting the missing units. We added them in the figure caption.

The liquid water path measurements are based on the microwave radiometer, while the observed liquid water content is based on the Cloudnet algorithm, that cominbes several instruments and model data to classify clouds. In this case, there are only a few areas in the cloud, that the algorithm classifies as liquid, hence the LWCs are very high. We added the following text in the manuscript (line 236): *"In Table 3, the observed LWP is smaller than the observed LWC. The LWP measurements are based on a microwave radiometer. The LWC is based on the Cloudnet algorithm, which combines several instruments and model data to classify the cloud. In this case, the areas for a liquid cloud occur only sporadically, leading to a very high LWC given the measured LWP. Hence, the interpretation of the LWC should be done with care."*.